# Self-promoted electroactive biomimetic mineralized scaffolds for bacteria-infected bone regeneration

Zixin Li[1,2,3,7], Danqing He[1,4,7], Bowen Guo[2], Zekun Wang[2], Huajie Yu[5], Yu Wang [1,4], Shanshan Jin[1,4], Min Yu[1,4], Lisha Zhu[1,4], Liyuan Chen[1,4], Chengye Ding[1,4], Xiaolan Wu[1,4], Tianhao Wu[1,4], Shiqiang Gong [6], Jing Mao[6], Yanheng Zhou[1,4], Dan Luo [2] ✉ & Yan Liu [1,4] ✉

Infected bone defects are a major challenge in orthopedic treatment. Native bone tissue possesses an endogenous electroactive interface that induces stem cell differentiation and inhibits bacterial adhesion and activity. However, traditional bone substitutes have difficulty in reconstructing the electrical environment of bone. In this study, we develop a self-promoted electroactive mineralized scaffold (sp-EMS) that generates weak currents via spontaneous electrochemical reactions to activate voltage-gated $Ca^{2+}$ channels, enhance adenosine triphosphate-induced actin remodeling, and ultimately achieve osteogenic differentiation of mesenchymal stem cells by activating the BMP2/ Smad5 pathway. Furthermore, we show that the electroactive interface provided by the sp-EMS inhibits bacterial adhesion and activity via electrochemical products and concomitantly generated reactive oxygen species. We find that the osteogenic and antibacterial dual functions of the sp-EMS depend on its self-promoting electrical stimulation. We demonstrate that in vivo, the sp-EMS achieves complete or nearly complete in situ infected bone healing, from a rat calvarial defect model with single bacterial infection, to a rabbit open alveolar bone defect model and a beagle dog vertical bone defect model with the complex oral bacterial microenvironment. This translational study demonstrates that the electroactive bone graft presents a promising therapeutic platform for complex defect repair.

Infected bone defects are considered a serious challenge in the field of orthopedics due to their high treatment failure rate[1,2]. Bacterial infection severely hinders bone regeneration. The open environment of bone defects in oral cavity is conducive to the colonization and reproduction of various opportunistic pathogens. Once a bacterial biofilm is formed, the repeated stimulation of infected tissues in the lesion results in osteoinflammatory hyperplasia and avascular necrosis of new bone, thereby greatly inhibiting the process of bone healing[3,4]. To date, bone grafting combined with antibiotics is a common strategy for the clinical treatment of infected bone defects[5,6]. However, it is difficult for antibiotics to deal with complex infections caused by drug-resistant bacteria. More seriously, antibiotics also inhibit the osteoinductive properties of bone grafts by disrupting the balance of the body's microbial ecosystem, ultimately leading to a "double loss" of anti-infection and bone repair[7,8]. This dilemma underscores the desperate need to develop novel bone tissue engineering strategies to balance antibacterial and osteoinductive properties.

Piezoelectricity of native bone has been identified as an essential factor in maintaining the physiological function of bone[9–11]. Bone tissue can spontaneously generate endogenous electrical signals, which are

derived from the polarization of collagen fibrils caused by mutual misalignment under the applied shear force[12,13]. These weak electrical signals play a decisive role in stem cell differentiation and tissue regeneration: electrical stimulation increases the influx of $Ca^{2+}$ by stimulating voltage-gated $Ca^{2+}$ channels (VGCCs) and activating the $Ca^{2+}$ signaling pathway to enhance stem cell osteogenic differentiation, and it can also promote cell metabolism and accelerate adenosine triphosphate (ATP) depletion, thereby achieving cytoskeleton reorganization and changing membrane-related cellular behaviors, such as endocytosis, exocytosis, adhesion, migration, and proliferation[14–16]. In addition, electrical stimulation has been proven to promote osteoblast attachment and proliferation while inhibiting bacterial activity[17]. It can be speculated that implanted scaffolds for bone tissue engineering should not only be similar to natural tissue in chemical properties and micro-nanotopology but should also simulate a natural-like electrical microenvironment to boost its inductive properties.

In light of these considerations, in this study we constructed a self-promoted electroactive mineralized scaffold (sp-EMS) via biomimetic self-assembly of mineralized collagen fibrils and their structural analogs, silver ultrathin nanowires (Ag uNWs), for the effective treatment of bacteria-infected bone defects. It has been demonstrated that the biomimetic self-assembly strategy can thermodynamically control the hierarchical arrangement of hydroxyapatite nanocrystals and collagen molecules to form a biomimetic mineralized collagen interface that is highly similar to natural bone in terms of mechanical properties, chemical composition, surface topology, biocompatibility, and biodegradability. This bone-like interface can regulate the fate of stem cells and exhibit excellent bone regeneration potential[18,19]. Inspired by this, further coassembly with electroactive nanomaterials will not only preserve the biomimetic bone-like interface, but also endow the scaffold with antibacterial properties and stronger osteoinductive capabilities. The sp-EMS continuously generated a weak current through a mild electrochemical reaction, which stimulated the recruitment of bone marrow mesenchymal stem cells (BMSCs) and promoted their osteogenic differentiation and angiogenesis. Meanwhile, electrical stimulation also synergized with the electrochemical oxidation product, silver ions, to destroy the cell wall of Staphylococcus aureus (S. aureus) and induce oxidative stress, resulting in bacterial death. Rat critical-sized noninfected and infected bone defect models were used to detect the bone regeneration performance and mechanism of the sp-EMS, confirming its osteogenic and antibacterial properties in vivo. In addition, we established complex infection bone defects for large animals including rabbits and dogs, to validate the clinical translational potential of the sp-EMS as a bone substitute material.

## Results

### Synthesis and characterization of self-promoted electroactive mineralized scaffolds

The sp-EMS was obtained by intrafibrillar mineralization of the coassembled Ag uNWs with collagen molecules (Fig. 1a)[19,20]. Transmission electron microscopy (TEM) demonstrated that the average diameter of Ag uNWs was ~19.20 ± 4.00 nm, while their lengths could reach tens of micrometers (Fig. 1b). Ultrahigh aspect ratio geometry endowed Ag uNWs with spatial configurational freedom. Generally, conventional metal nanowires with thicker diameters are structurally rigid; however, the synthesized ultrahigh aspect ratio Ag uNWs possessed a certain flexibility and bent on the carbon-supported film, as observed by TEM. Furthermore, atomic force microscopy (AFM) showed that the Young's modulus of Ag uNWs ranged from 0.22 to 0.59 GPa (Supplementary Fig. 1), which was similar to that of collagen microfibrils[21]. In addition, the Ag uNWs were highly similar to the microfibrils formed during collagen self-assembly in both geometrical features and flexible configurations, which is the basis for the subsequent successful coassembly of the two materials. As shown by TEM, the coassembled sp-EMS fibrils exhibited a periodic light- and dark-staggered

nanostructure with D-periods of 63.9 ± 3.3 nm, consistent with native mineralized collagen[19,20]. More importantly, Ag uNWs with higher contrast were distributed in the intrafibrillar region along the extension direction of sp-EMS fibrils (Fig. 1c). In contrast, conventional metal nanowires with larger diameter were structurally rigid, which possessed a straight conformation without any bending on TEM substrates and were difficult to coassemble with mineralized collagen fibrils (Supplementary Fig. 2a). Due to the mismatch of geometric shapes and spatial configurational freedom, the thick Ag NWs were prone to phase separation from the collagen phase or directly penetrated the fibrous structure, resulting in the failure of coassembly (Supplementary Fig. 2b).

To further confirm the successful assembly of Ag uNWs into mineralized collagen fibrils, the elemental distribution of the sp-EMS fibril was investigated by STEM-coupled energy-dispersive X-ray spectroscopy (EDS) mapping (Fig. 1d). The uniform distribution of nitrogen, phosphorus, and calcium confirmed that sp-EMS fibrils showed a precise mineralized structure, while silver was mainly concentrated in the high-contrast nanowire-like structure inside the sp-EMS fibril and extended along the fibril, which was consistent with the TEM results. After ultrasonic-assisted extraction, inductively coupled plasma optical emission spectroscopy (ICP-OES) results confirmed that the mass content of Ag in the sp-EMS was 4.81%, which was close to the inventory ratio of reactants (Fig. 1e). X-ray diffraction analysis (XRD) also confirmed that the sp-EMS yielded both $Ag^0$ and hydroxyapatite lattice parameters (Fig. 1f). Thermogravimetric analysis (TGA) further provided direct evidence to determine the mass fraction of Ag in the sp-EMS (Fig. 1g). Both the sp-EMS and mineralized self-assembled collagen scaffolds (MS) possessed almost identical thermal-induced weight loss trends: mass fraction loss in the temperature range of 50–120 °C was associated with the loss of adsorbed or weakly bonded water molecules; the sharp drop in weight from 200 to 600 °C was attributed to dehydration due to collagen carbonization. Due to the extremely high boiling point of silver, the difference in residual mass between the sp-EMS and MS could be attributed to the content of coassembled Ag uNWs (4.96%), which was consistent with the ICP results. The entire coassembled system was maintained by weak forces, which did not involve the formation and breaking of chemical bonds or change the coordination state of the silver nanostructures. As shown by X-ray photoelectron spectroscopy (XPS), peaks corresponding to C, N, O, P, Ca, and Ag appeared in the spectrum of the sp-EMS (Fig. 1h), while no Ag peak was observed in the spectrum of the MS group (Supplementary Fig. 3). A detailed scan of the Ag region showed that the sp-EMS contained a dual peak at binding energies of 374.6 eV and 366.8 eV, corresponding to the Ag ($3d_{3/2}$) and ($3d_{5/2}$) of metallic Ag, respectively, consistent with Ag uNWs (Fig. 1i). The surface topology and nanomechanical properties of the sp-EMS were further determined by AFM (Fig. 1j). Similar to native mineralized collagen fibrils, the surface of sp-EMS fibrils follows an undulating topography with a periodic bisignate peak-dip shape at its cross-sectional height along the z-axis, which further confirmed that the Ag uNWs were assembled into the interior of the mineralized collagen fibril matrix rather than deposited on the surface. Coassembly of Ag uNWs hardly affected the mechanical properties of mineralized collagen. As shown by the AFM results, the Young's modulus of sp-EMS fibrils ranged from 9.01 to 13.99 GPa, which was not significantly different from that of the MS group. Through bottom-up assembly methodology, sp-EMS fibrils can be further assembled into centimeter-scale 3D porous implantable scaffolds with a high porosity of 91.09 ± 1.28% and interconnected pores 117.12 ± 25.87 μm in size (Fig. 1k, l), which provide a high surface area with a bone-like microenvironment to promote recruitment of stem cells and facilitate subsequent stem cell differentiation via self-promoted electrical stimulation. Under physiological conditions, the sp-EMS could self-promote the sustained release of weak electric current, mainly attributed to the mild electrochemical corrosion reaction of Ag uNWs in solution and the concomitant ionic

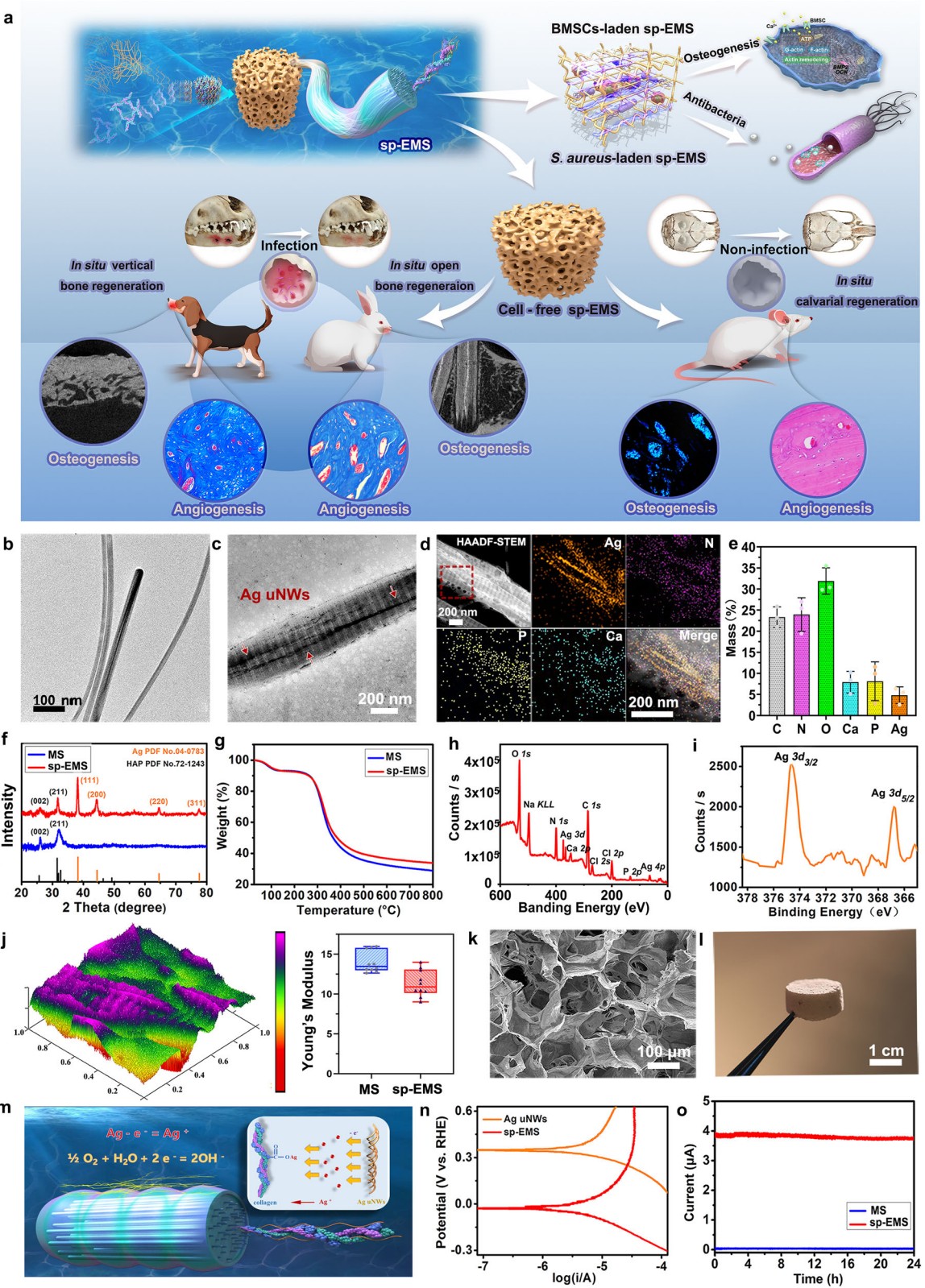

displacement in the assembled scaffold. The electrochemical reaction equations were inferred as follows:

$$Ag - e^- = Ag^+$$

$$1/2\,O_2 + H_2O + 2e^- = 2\,OH^-$$

Generally, the electrochemical corrosion reaction of Ag nanostructures is more difficult to occur, attributed to their higher corrosion potential. However, the carboxyl-rich structures enriched in the collagen molecules of sp-EMS were prone to coordinate with Ag ions (Supplementary Fig. 4), the corrosion products of Ag, which promoted the electrochemical corrosion and the ionic displacement (Fig. 1m). As shown in the Tafel curve (Fig. 1n), the corrosion potential of sp-EMS

**Fig. 1 | Physicochemical properties of the sp-EMS: micro-nano topology and self-promoted electrical performance. a** Design and synthesis of the sp-EMS for in situ regeneration of bone defects in rats, rabbits, and dogs. **b** A representative TEM image of Ag uNWs. **c** A representative unstained TEM image of sp-EMS fibril with hierarchical deposition of hydroxyapatite nanocrystals to form native bone-like D-periodic nanostructures ($D \approx 67$ nm). Ag uNWs with higher contrast were distributed inside the sp-EMS fibril. **d** Energy-dispersive X-ray spectroscopy elemental map of the sp-EMS fibril, including the representative HAADF-STEM image, mappings of silver/nitrogen/phosphorus elements, and the merged photo. **e** Mass content of Ag, P, C, N, O, and Ca in sp-EMS measured by ICP-OES ($n = 3$ biologically independent samples. Data are presented as means ± SD). **f** XRD patterns of the MS and the sp-EMS compared with that of standard hydroxyapatite and silver nanocrystals. **g** TGA of the MS and the sp-EMS. **h** XPS measurements of the sp-EMS. **i** high-resolution XPS spectra of Ag (*3d*). **j** Representative 3D topology of the sp-EMS fibril measured by AFM (left) and Young's modulus of the MS and the sp-EMS (right) ($n = 10$ biologically independent samples, by two-tailed Student's *t*-test: $P > 0.05$. Each box plot represents the minimum, first quartile, median, third quartile, and maximum of 10 values; the whiskers were drawn down to the minimum and up to the maximum). **k, l** A representative SEM image and gross morphology of 3D sp-EMS, respectively. **m** Schematic illustration of the mechanism on self-promoted electrogenesis by the sp-EMS in a physiological environment. **n** Tafel curve of sp-EMS and Ag uNWs. **o** Microcurrents generated on the surface of the MS and the sp-EMS in PBS buffer. All experiments were repeated independently at least three times. Source data and exact *P* values are provided in the Source Data file.

($-0.03$ V vs. RHE) was much lower than that of Ag uNWs (0.35 V vs. RHE), confirming that the assembly of Ag uNWs and collagen in sp-EMS accelerated the corrosion reaction[22]. In phosphate buffered saline (PBS), the sp-EMS produced a stable current with a magnitude of ~ 4.0 μA; in contrast, the MS did not produce any electrical response due to the lack of Ag nanostructures[22] (Fig. 1o).

## Self-promoted electrostimulation induces osteogenic differentiation of mesenchymal stem cells via calcium channels

Electrical stimulation, as a biophysical cue, can effectively manipulate cell behaviors, including cell morphology, growth and osteogenesis[23]. To elucidate the osteoinductive potential of the sp-EMS via self-promoted electrical stimulation, BMSCs were seeded on different scaffolds, including the sp-EMS and electrically inactive MS, as a control. Cells seeded in 6-well plates served as a blank group. After 24 h of culture, 98.95 ± 1.02% of BMSCs were alive in the sp-EMS group, comparable to the MS (97.22 ± 3.05%) and blank (97.02 ± 1.39%) groups (Supplementary Fig. 5a). The Cell Counting Kit (CCK)-8 assay showed that the sp-EMS promoted BMSC proliferation from day 1 to day 7, similar to the MS and blank groups. Thus, the sp-EMS possessed good biocompatibility (Supplementary Fig. 5b).

Cellular electrical responses to electroactive scaffolds originate from transmembrane ion flow and transfer. In particular, electrical stimulation modulates the inflow and outflow of Ca$^{2+}$, thereby altering cell morphology and affecting cell function[24]. Herein, the sp-EMS promoted the influx of calcium ions in BMSCs, as evidenced by higher expression of the calcium fluorescent probe Fluo-3 in the sp-EMS group than in the MS and blank groups ($P < 0.001$, Fig. 2b; Supplementary Fig. 6a). The significantly elevated calcium influx levels were attributed to the activation of voltage-dependent calcium channel α2/δ subunit 1 (CACNA2D1) in BMSCs by the sp-EMS. The mRNA expression level of *CACNA2D1* in BMSCs was significantly upregulated in the sp-EMS group at both day 7 and day 14 compared to the MS group ($P < 0.001$, Fig. 2c). As an energy source for life activities, ATP can initiate a signaling cascade that enhances osteogenesis. Electrical stimulation can directly guide the migration of protons to the mitochondrial membrane and bind with ATPase to promote ATP generation; in addition, electrical stimulation-induced calcium influx will further assist ATP synthesis by activating Ca$^{2+}$-sensitive dehydrogenase and ATP synthase[16]. To further explore the effect of the sp-EMS on cellular energy metabolism, the presence of purinoceptor (P2X7), an important indicator of ATP content, was measured on days 7 and 14. The mRNA expression level of *P2X7* in BMSCs was significantly upregulated in the sp-EMS group at day 7 and day 14 ($P < 0.001$ vs. MS and blank), confirming that the sp-EMS promoted ATP synthesis. In contrast, the mRNA expression of *P2X7* in the MS group was not significantly different from that in the blank group ($P < 0.05$, Fig. 2c). The generated ATP will promote the conversion of G-actin to F-actin, thereby realizing the remodeling and reorganization of the cytoskeleton, which lays the foundation for the differentiation of stem cells into osteoblasts. Both the F-actin protein level and the F-actin/G-actin ratio were significantly increased in the sp-EMS group compared with

the MS and blank groups ($P < 0.001$, Fig. 2d). The actin cytoskeleton network plays an important role during bone morphogenetic protein 2 (BMP2)-induced osteoblast differentiation. Cytoskeletal reorganization drives mesenchymal condensation and regulates downstream molecular signaling, including BMPs[25]. Real-time PCR showed that after 7 days and 14 days of coculture, the expression levels of the BMSC osteogenesis-associated markers *BMP2* ($P < 0.001$ vs. blank and $P < 0.01$ vs. MS) and *OCN* ($P < 0.001$ vs. blank and MS groups) were significantly upregulated in the sp-EMS group (Fig. 2e). Smad1/5 is involved in the canonical BMP signaling pathway[26]; therefore, the expression of BMP2 and Smad1/5 was investigated to reveal the specific mechanism of BMSC osteogenesis mediated by the sp-EMS. Western blotting showed that the protein expression levels of BMP2 and p-Smad1/5/9 were significantly increased in the sp-EMS group compared to the blank and MS groups ($P < 0.001$, Fig. 2f; Supplementary Fig. 6c). These results suggest that self-promoted electrical stimulation in sp-EMS promoted calcium influx and enabled actin remodeling of BMSCs by increasing ATP content, ultimately enhancing osteogenic differentiation through activation of the BMP2/Smad5 pathway (Fig. 2a).

To confirm that the osteogenic induction of BMSCs originates from the electrical stimulation produced by the sp-EMS, rather than its chemical/micronano structure, AgCl and Ag$_2$S structures were formed on the surface of Ag uNWs by passivation to suppress the electrochemical reaction. Cyclic voltammetry curves showed that the current on the surface of the mineralized collagen/AgCl (MS/AgCl) and mineralized collagen/Ag$_2$S (MS/Ag$_2$S) coassembled scaffolds almost disappeared (Fig. 2g). Scaffolds lacking self-promoted electrogenic capacity directly downregulated calcium influx from BMSCs seeded on their surfaces, and Fluo-3-positive cells were significantly reduced in the MS/AgCl and MS/Ag$_2$S groups compared with the sp-EMS group ($P < 0.001$, Fig. 2h; Supplementary Fig. 6b). Consistently, the mRNA expression of the calcium channel *CACNA2D1* ($P < 0.001$ at Day 7 and 14 vs. sp-EMS) and purinoceptor *P2X7* ($P < 0.001$ at Day 7 and 14 vs. sp-EMS) was repressed after passivation (Fig. 2i), and F-actin/G-actin was also decreased in the MS/AgCl and MS/Ag$_2$S groups ($P < 0.001$, Fig. 2j). Moreover, BMSC osteogenic differentiation was significantly inhibited, as evidenced by decreased mRNA expression of *BMP2* and *OCN* ($P < 0.001$, Fig. 2k). Western blotting analysis also showed the downregulation of BMP2 and p-Smad1/5/9 expression in the MS/Ag$_2$S and MS/AgCl groups, indicating that the passivated scaffolds (MS/AgCl and MS/Ag$_2$S) restricted the activation of the BMP2/Smad5 pathway ($P < 0.001$, Fig. 2l; Supplementary Fig. 6d).

Typically, tissue engineering uses an external power source or battery to apply electrical stimulation. To further verify that the sp-EMS has the equivalent electrical stimulation ability to realize the osteogenesis-promoting function, we used a signal generator to directly input a current of 4 μA to the BMSC medium through the electrode, and observed the biochemical reaction of the cells after 48 h of electrical stimulation (Supplementary Fig. 7a). Direct electrical stimulation, similar to that provided by the sp-EMS, induced calcium influx (Supplementary Fig. 7b), upregulated the mRNA expression of

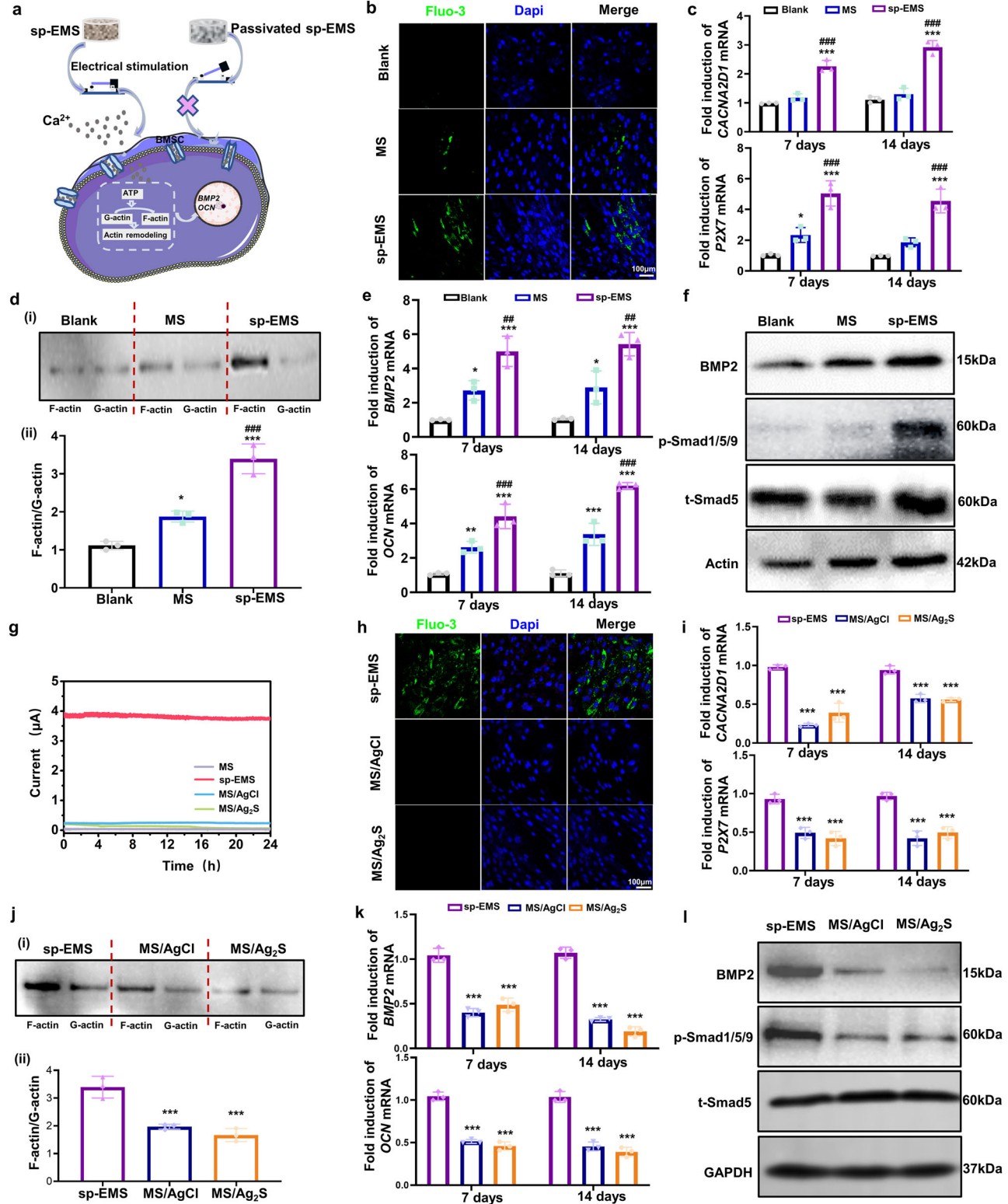

CACNA2D1 and P2X7 (P < 0.05 and P < 0.001, respectively; Supplementary Fig. 7c), and increased the expression of F-actin and the ratio of F-actin/G-actin (P < 0.001; Supplementary Fig. 7d). The osteogenic differentiation of BMSCs induced by direct electrical stimulation was consistent with that of the sp-EMS group: the mRNA expression levels of BMP2 and OCN increased significantly (P < 0.01 and P < 0.05, respectively; Supplementary Fig. 7e), while BMP2 and p-smad1/5/9 protein expression levels were also upregulated (Supplementary Fig. 7f). The above results consistently confirm that the sp-EMS can

generate electrical stimulation equivalent to an external power source through a self-promoted electrochemical reaction.

## The sp-EMS completely regenerates rat critical-sized non-infected calvarial defects in situ

In native bone tissues, electrical signals generated by shear–stressed collagen and deformation of fluid-filled channels play important roles in bone remodeling[12]. Therefore, self-promoted electrical stimulation in the sp-EMS may restore the electrical environment of the defective

**Fig. 2 | Self-promoted electrostimulation induces osteogenic differentiation of mesenchymal stem cells via calcium channels. a** Schematic of osteogenic differentiation of BMSCs induced by the sp-EMS. **b** Representative immunofluorescence images of Fluo-3 (green) in BMSCs seeded on the 6-well plate (blank), the MS, and the sp-EMS for 7 days. **c** Relative mRNA expressions of calcium channel (*CACNA2D1*) and purinoceptors (*P2X7*) in BMSCs seeded on different groups for 7 and 14 days (*n* = 3 biologically independent samples, by one-way ANOVA with Tukey's post hoc test: \*\*\**P* < 0.001 versus blank; ### *P* < 0.001 versus MS). **d** (i) Western blotting of actin in BMSCs seeded on different scaffolds for 7 days by differential ultracentrifugation. (ii) Ratio of F-actin to G-actin from densitometry of actin bands in (i) (*n* = 3 biologically independent samples, by one-way ANOVA with Tukey's post hoc test: \**P* < 0.05, \*\*\**P* < 0.001 versus blank; ###*P* < 0.001 versus MS). **e** Relative mRNA expressions of the osteogenic differentiation markers (*BMP2* and *OCN*) in BMSCs cultured on different scaffolds for 7 and 14 days (*n* = 3 biologically independent samples, by one-way ANOVA with Tukey's post hoc test: \**P* < 0.05, \*\**P* < 0.01, \*\*\**P* < 0.001 versus blank; ##*P* < 0.01, ###*P* < 0.001 versus MS). **f** Western blotting of the expressions of BMP2, p-Smad1/5/9, and t-Smad5 in BMSCs cultured on different scaffolds for 7 days. Beta-actin served as an internal control for equal loading. **g** The electrical signal of the passivated sp-EMS. **h** Representative immunofluorescence images of Fluo-3 (green) in BMSCs seeded on the sp-EMS and passivated sp-EMS (MS/AgCl and MS/Ag$_2$S scaffolds) for 7 days. **i** Relative mRNA expressions of *CACNA2D1* and *P2X7* in BMSCs cultured on the sp-EMS, MS/AgCl and MS/Ag$_2$S for 7 and 14 days (*n* = 3 biologically independent samples, by one-way ANOVA with Tukey's post hoc test: \*\*\**P* < 0.001 versus sp-EMS). **j** (i) Western blotting of actin in BMSCs seeded on different scaffolds for 7 days by differential ultracentrifugation. (ii) Ratio of F-actin to G-actin from densitometry of actin bands in (i) (*n* = 3 biologically independent samples, by one-way ANOVA with Tukey's post hoc test: \*\*\**P* < 0.001 versus sp-EMS). **k** Relative mRNA expressions of *BMP2* and *OCN* in BMSCs seeded on the sp-EMS, MS/AgCl, and MS/Ag$_2$S for 7 and 14 days (*n* = 3 biologically independent samples, by one-way ANOVA with Tukey's post hoc test: \*\*\**P* < 0.001 versus sp-EMS). **l** Western blotting of the expressions of BMP2, p-Smad1/5/9, and t-Smad5 in BMSCs cultured on different scaffolds for 7 days. GAPDH served as an internal control for equal loading. All experiments were repeated independently at least three times. Data are presented as means ± SD. Source data and exact *P* values are provided in the Source Data file.

tissue and has the potential to accelerate bone regeneration. To test this hypothesis, sp-EMS was transplanted into rat critical-sized calvarial defects without the use of exogenous cells and growth factors, while implantation of the electrically inactive MS was set as a control group, and the defect without scaffold implantation served as a blank group (Fig. 3a).

The engineered tissues at 8 weeks and 12 weeks after implantation were first characterized by micro computed tomography (µCT). After 8 weeks of implantation, most defect areas, including the defect center, were filled with new bone in the sp-EMS group (77.45 ± 2.88%), which was much more than that in the MS (31.55 ± 8.22%, *P* < 0.001) and blank groups (6.17 ± 2.53%, *P* < 0.001). The same tendency could be observed in the changes in bone volume. The bone volume in the sp-EMS group (60.80 ± 2.26 mm³) was higher than that in the MS group (24.77 ± 6.45 mm³, *P* < 0.001) and the blank group (4.84 ± 1.99 mm³, *P* < 0.001; Supplementary Fig. 8a). The sagittal and coronary sections of the defect area also showed higher new bone areas and increased bone volume after sp-EMS implantation. Similar to the µCT results, hematoxylin-eosin (HE) staining also revealed that the defect area was mostly filled with bone-like structures, including the defect center in the sp-EMS group. In contrast, large amounts of remnant scaffolds were detected in the MS group, and almost no obvious neo-bone was formed in the blank group (Supplementary Fig. 8b). As the implantation time extended to 12 weeks, the 3D reconstructed µCT images showed that the calvarial bone defects were almost healed (96.93 ± 6.22%) by the sp-EMS, including the defect center, whereas 67.74 ± 5.77% of the defect area was repaired in the MS group, and little new bone was formed (12.24 ± 4.14%) along the defect margin in the blank group (*P* < 0.001 vs. blank and MS groups). In addition, the bone volume of the sp-EMS group (76.12 ± 4.89 mm³) was significantly increased compared with that of the MS group (54.48 ± 5.63 mm³) and the blank group (9.85 ± 3.04 mm³, *P* < 0.001) (Fig. 3b). Sagittal and coronary sections further demonstrated that both ends of the bone defects were continuous in the sp-EMS group, while the defect center was discontinuous in the MS and blank groups (Fig. 3c). Histological analysis of the engineered tissues was consistent with the µCT results. In the sp-EMS group, dense and native bone-like structures with blood vessels and osteocytes/osteoblasts filled the defect area, even at the defect center, whereas only some sparse neo-bone formed in the MS group. Moreover, the sp-EMS was almost degraded, and Ag could not be detected by SEM mapping in the sp-EMS-regenerated new bone (Supplementary Fig. 9a), while some remnant scaffolds could still be observed in the defect area of the MS group. In the blank group, soft tissues filled the defect area, and almost no obvious new bone formation was observed (Fig. 3c). HE staining of major metabolic organs after 2 weeks and 12 weeks of implantation showed that the sp-EMS showed no significant toxicity to liver and kidney (Supplementary

Fig. 9c and d). The silver ions released by the sp-EMS could be eliminated from the body through urine and feces (Supplementary Fig. 9b)[27]. ICP-OES further confirmed that almost no residual silver remained in major metabolic organs at 12 weeks after implantation, consistent with the silver levels in unexposed subjects (Supplementary Fig. 9b)[28]. These findings indicate that the sp-EMS possesses strong osteoinductive ability and biosafety, and achieves an optimal balance between bone remodeling and scaffold degradation.

The strong osteoinductive ability of sp-EMS was attributed to the continuous electrical stimulation after implantation. To confirm that electrical stimulation accompanies the bone regeneration process, we measured potentials and currents of sp-EMS in situ at different periods after implantation in a rat critical-sized noninfected calvarial defect model (Supplementary Movie 1). After 24 h of implantation, the microcurrent generated on the surface of the sp-EMS was ~4.5 µA, similar to the value measured in phosphate buffered saline (~4.0 µA). Over time, the microcurrent intensity on the sp-EMS surface decreased to ~4.0 µA on day 3, ~3.0 µA on day 7, ~2.0 µA on day 14, and ~0.02 µA on day 21 respectively (Fig. 3d). To better understand the mechanisms underlying the bone-promoting effect of the sp-EMS, the electrical responsiveness and downstream molecular pathways of host stem cells were further examined using immunofluorescence and immunohistochemical staining. The voltage-dependent calcium channel CACNA2D1 was further characterized by immunofluorescence staining in the engineered tissue after 2 weeks of implantation. A higher expression level of CACNA2D1 was detected in the defect areas in the sp-EMS group than in the MS and blank groups. The number of CACNA2D1-positive cells in the defect area of the sp-EMS group (18.02 ± 4.32%) was much higher than that in the MS (2.28 ± 0.87%) and blank groups (0.00 ± 0.01%, *P* < 0.001) (Fig. 3e; Supplementary Fig. 10). Generally, neobone formation requires a coordinated process of angiogenesis and osteogenesis at the repair site. To investigate whether the sp-EMS activates CACNA2D1 through electrical stimulation to promote the secretion of autocrine and paracrine factors from endothelial cells and osteoblasts, the engineered tissue was further characterized for vascularization and osteogenesis assessment by immunochemistry (Fig. 3f). Increased expression levels of the osteogenesis-associated markers BMP2 and osteocalcin (OCN) and the vascularization-related factors vascular endothelial growth factor receptor-1 (VEGFR-1) and platelet endothelial cell adhesion molecule-1 (PECAM-1/CD31) were detected in the sp-EMS group. Semiquantitative analysis showed that the numbers of BMP2+ cells, OCN+ cells, VEGFR-1+ cells, and CD31+ cells were all increased in the sp-EMS group compared with the MS group and blank group (*P* < 0.01, Supplementary Fig. 11). These results suggest that the sp-EMS exhibits dual functions of osteoinduction and angioinduction, and accelerates endogenous bone regeneration.

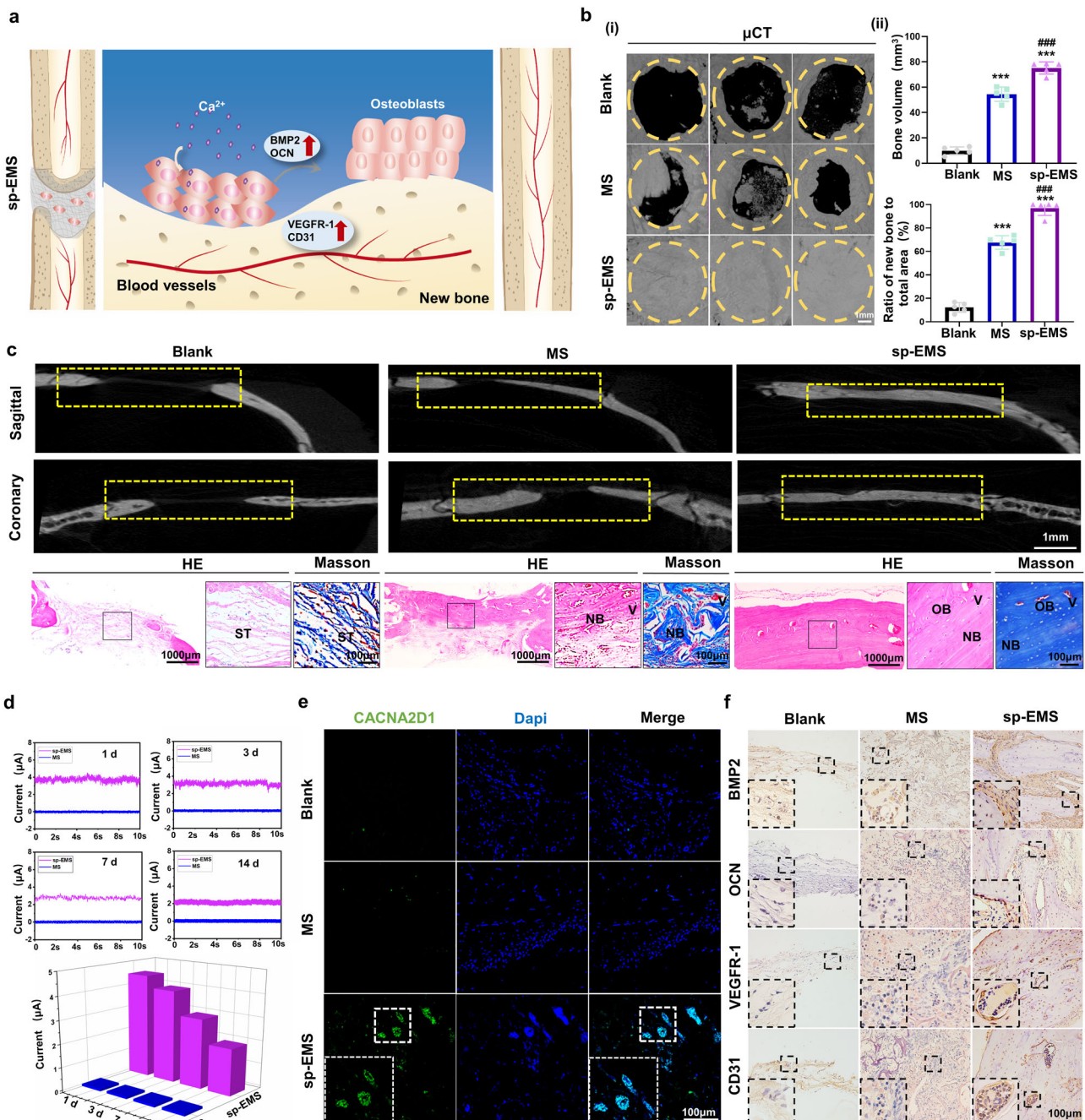

**Fig. 3 | The sp-EMS completely regenerates rat critical-sized noninfected calvarial defects in situ. a** Schematic of the healing process of rat noninfected calvarial bone defects after the cell-free sp-EMS implantation. **b** (i) Representative μCT images of rat noninfected calvarial bones in the blank, MS, and sp-EMS groups after 12-week implantation. Yellow circles display defect boundary. (ii) Semiquantification of the bone volume and the ratio of new bone to total area in (i) ($n = 5$ rat critical-sized noninfected calvarial defects per group, by one-way ANOVA with Tukey's post hoc test: \*\*\*$P < 0.001$ versus blank; ###$P < 0.001$ versus MS). **c** Representative μCT, HE and Masson's trichrome staining images of the non-infected engineered bones in different groups after 12-week implantation. ST: soft tissue, NB: new bone, V: vessel, OB: osteoblasts. Yellow boxed areas show the defect areas. **d** Microcurrents generated on the surface of the MS and the sp-EMS in rat critical-sized noninfected calvarial defects on days 1, 3, 7, and 14. **e** Representative immunofluorescence images of CACNA2D1 (green) in defect areas after 2 weeks of implantation. **f** Representative immunohistochemical staining images of BMP2, OCN, VEGFR-1, and CD31 in newly formed calvaria areas after 8 weeks of implantation. All experiments were repeated independently at least three times. Data are presented as means ± SD. Source data and exact $P$ values are provided in the Source Data file.

## The sp-EMS potentiates the bactericidal effects by self-promoted electrostimulation in vitro

Natural bone can form microarea electrostatic fields under stress conditions, thereby inhibiting the adhesion and proliferation of bacteria[29]. Different from traditional electrically inert tissue-engineering scaffolds, the sp-EMS not only simulates the native bone electrical microenvironment through self-promoted electrochemical reactions but also releases electrochemical reaction products to achieve antibacterial properties (Fig. 4a). First, the sp-EMS inhibited bacterial growth, as evidenced by markedly decreased *S. aureus* colonies (Fig. 4b). Second, most *S. aureus* exposed to the sp-EMS was dead (74.50 ± 4.96% bacteria stained with red), while bacteria in the MS

(96.21 ± 0.91% bacteria stained with green) and blank groups (98.31 ± 1.05% bacteria stained with green) were alive, as determined by live/dead staining. (Supplementary Fig. 12a, b). Finally, the sp-EMS influenced the micro-nanostructure of *S. aureus*. SEM showed that bacterial adhesion was almost inhibited when treated with the sp-EMS, whereas smooth and intact morphology of *S. aureus* was still observed in the blank and MS groups after 24 h of incubation. TEM revealed that the sp-EMS completely disrupted the cell membrane of *S. aureus*, whereas intact bacterial nanostructures were observed in the MS and blank groups (Fig. 4b). Furthermore, the antibacterial effect of the sp-EMS was blocked by the loss of self-prompted electrical stimulation when the silver nanowires were passivated (*P* < 0.001; Fig. 4c).

The excellent antibacterial properties of the sp-EMS may be derived from the intermediate product of Ag$^+$ and the accompanying reactive oxygen species (ROS) generated by self-promoted electrical stimulation, which can damage bacterial biomacromolecules (including lipids, proteins, and nucleic acids), reduce dehydrogenase activity, and induce bacterial oxidative stress, thereby preventing biofilm formation and leading to bacterial death[30]. To verify this hypothesis on the anti-infection mechanism, we further investigated the Ag$^+$ concentration and ROS content released by the sp-EMS. The Ag$^+$ release curve showed that the Ag$^+$ concentration of the sp-EMS gradually increased with time, which was attributed to the self-promoted electrochemical reaction of

the sp-EMS enabling the continuous conversion of Ag uNWs to Ag$^+$ (Fig. 4d). The generation of ROS by the sp-EMS was assessed by using an oxidation-sensitive fluorescent probe dye, 2′,7′-dichlorodihydro-fluorescein diacetate (DCF-DA). A fluorescence spectrophotometer showed that the ROS generated by the sp-EMS was 2.23 times greater than that in the MS group and 3.29 times greater than that in the blank group at 6 h (*P* < 0.001). The amount of ROS in the sp-EMS gradually increased with time, whereas there were almost no changes in the MS and blank groups (Fig. 4e). It was thus determined that the sp-EMS continuously released Ag$^+$ and produced ROS to inhibit bacterial activity.

Intracellular infection with *S. aureus* increases chemokine production by osteocytes, which may increase the release of the inflammatory cytokine TNF-α, thereby altering osteocyte function and survival. To elucidate whether the sp-EMS can inhibit infection-induced osteonecrosis, BMSCs were cultured in osteogenic induction medium conditioned with 10 µg/mL *S. aureus*-derived lipoteichoic acid (LTA), and the expression of osteogenesis and inflammation-related genes was investigated. Compared with the blank and MS groups, the mRNA expression levels of the inflammatory markers *TNF-α* and *iNOS* in BMSCs were downregulated on days 7 and 14 in the sp-EMS group. Conversely, the sp-EMS significantly enhanced the mRNA expression levels of the osteogenesis-related genes *BMP2* and *OCN* in BMSCs on days 7 and 14. These data indicate that the sp-EMS promoted

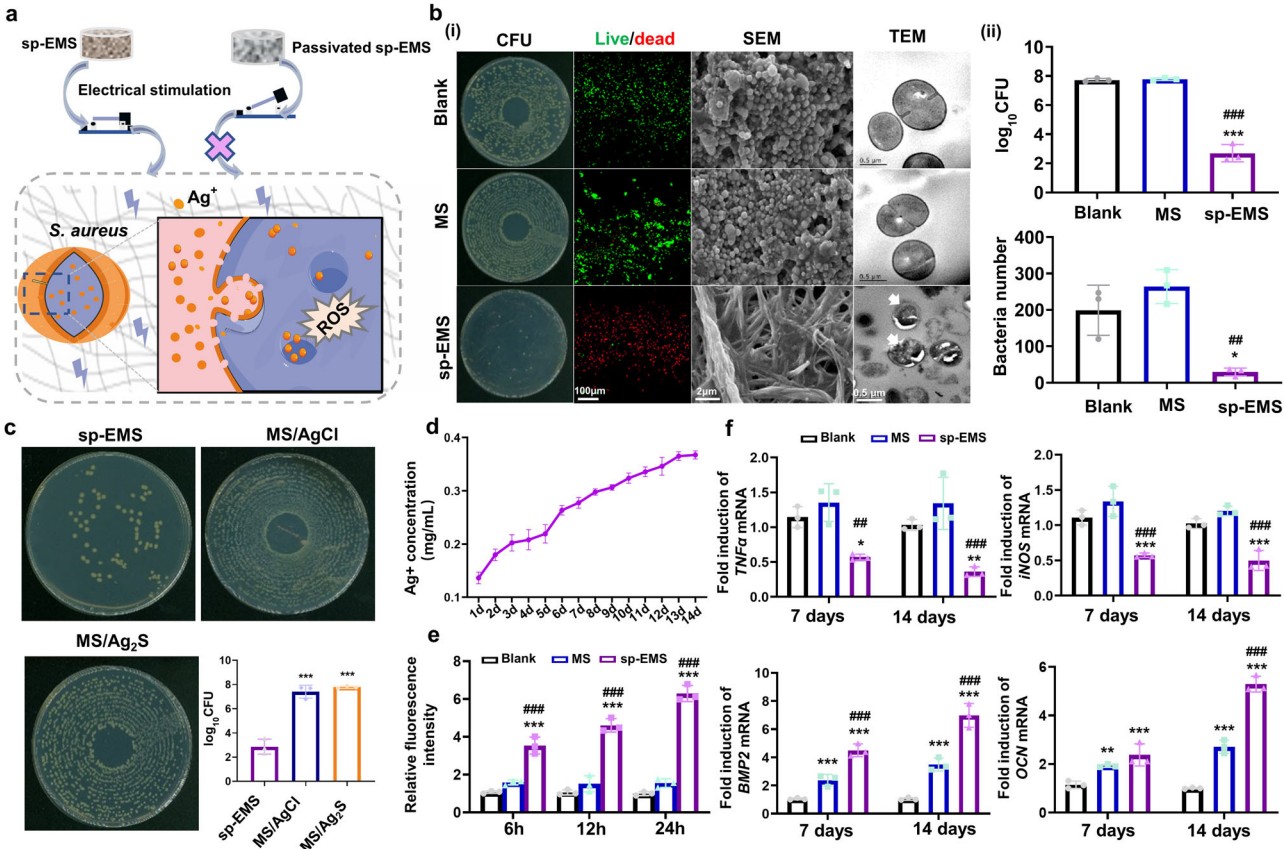

**Fig. 4 | The sp-EMS potentiates the bactericidal effects by self-promoted electrostimulation in vitro. a** Schematic diagram of the antibacterial mechanism of the sp-EMS. **b** (i) Representative CFU counting, live/dead staining, SEM, and TEM images of *S. aureus* seeded on the 6-well plate (blank), MS, and sp-EMS for 1 day. White arrows display the destruction of the cell membranes. (ii) Semiquantification of CFU counting and the bacteria number in SEM images in (i) (*n* = 3 biologically independent samples, by one-way ANOVA with Tukey's post hoc test: *\*P* < 0.05, \*\*\**P* < 0.001 versus blank; ##*P* < 0.01, ###*P* < 0.001 versus MS). **c** Representative CFU counting of *S. aureus* seeded on the sp-EMS, MS/AgCl, and MS/Ag$_2$S for 1 day (*n* = 3 biologically independent samples, by one-way ANOVA with Tukey's post hoc test:

\*\*\**P* < 0.001 versus sp-EMS). **d** The release curve of Ag$^+$ for 14 days. **e** Endogenous ROS level of *S. aureus* seeded on different scaffolds for 6 h, 12 h, and 24 h (*n* = 3 biologically independent samples). **f** Relative mRNA expressions of inflammatory-related markers (*TNF-α* and *iNOS*) and osteogenic differentiation markers (*BMP2* and *OCN*) in BMSCs cultured on different scaffolds for 7 and 14 days (*n* = 3 biologically independent samples, by one-way ANOVA with Tukey's post hoc test: \**P* < 0.005, \*\**P* < 0.01, \*\*\**P* < 0.001 versus blank; ##*P* < 0.01, ###*P* < 0.001 versus MS). All experiments were repeated independently at least three times. Data are presented as means ± SD. Source data and exact *P* values are provided in the Source Data file.

osteogenic differentiation of BMSCs under inflammatory conditions (Fig. 4f).

### The sp-EMS achieves nearly complete in situ bone regeneration in rats with single bacterial infection

To verify the osteogenic and antibacterial abilities of the sp-EMS, a rat bacteria-infected bone defect model was established (Fig. 5a). Since *S. aureus* accounts for 80–90% of the pathogenic bacteria of suppurative infected bone defects[6], *S. aureus* suspensions (100 μL, $1 \times 10^8$ CFU/mL) were applied to the calvarial bone defects, and then the sp-EMS and MS were transplanted to the defect areas. Infected bone defects without scaffold transplantation were set as blank groups. Compared to the rat calvarial noninfected bone defect model, the infected bone defect model without any implants exhibited a large accumulation of inflammatory cells 2 weeks after surgery, confirming the successful establishment of infection (Supplementary Fig. 13). After implantation for 12 weeks, as expected, the μCT results showed that the calvarial bone defects were almost repaired by the sp-EMS, including the defect center, whereas only less than half of the defect area was occupied by the new bone in the MS group, and the defect center was almost empty. Little new bone formed along the defect margin in the blank group (Fig. 5b). Semiquantitative analysis showed that the new bone area to total area in the sp-EMS group (89.16 ± 11.16%) was much higher than that in the MS (32.64 ± 5.29%) and blank groups (6.06 ± 1.68%, $P < 0.001$). Meanwhile, the new bone volume in the sp-EMS group (70.01 ± 8.74 mm³) was also much higher than that in the blank

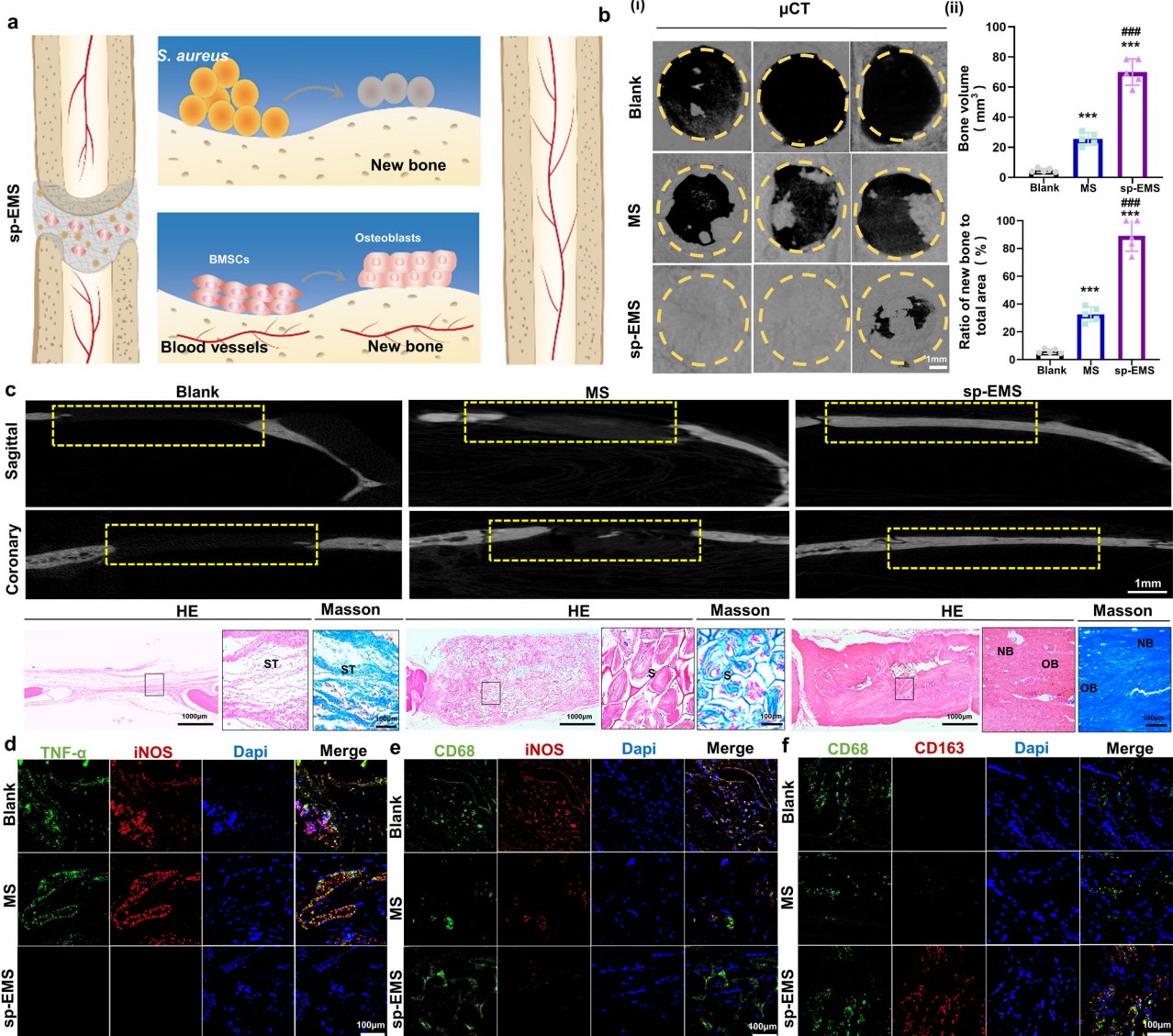

**Fig. 5 | The sp-EMS achieves nearly complete in situ bone regeneration in rats with single bacterial infection. a** Schematic of the healing process of rat infected calvarial bone defects after the cell-free sp-EMS implantation. **b** (i) Representative μCT images of rat infectious calvarial bones in different groups after 12-week implantation. Yellow circles display defect boundary. (ii) Semiquantification of the bone volume and the ratio of new bone to total areas in (i) (*n* = 5 rat critical-sized infected calvarial defects per group, by one-way ANOVA with Tukey's post hoc test: \*\*\**P* < 0.001 versus blank; ###*P* < 0.001 versus MS). **c** Representative μCT, HE, and Masson's trichrome staining images of the infected engineered bones in different groups after 12-week implantation. Yellow boxed areas show the defect areas. Black boxed areas show high-magnification views. ST: soft tissue, S: scaffolds, NB: new bone, V: vessels, OB: osteoblasts. **d** Representative immunofluorescence images of TNF-α staining (green) and iNOS staining (red) in defect areas after 2 weeks of implantation. **e** Representative immunofluorescence images of CD68 staining (green) and iNOS staining (red) in defect areas after 2 weeks of implantation. **f** Representative immunofluorescence images of CD68 staining (green) and CD163 staining (red) in defect areas after 2 weeks of implantation. All experiments were repeated independently at least three times. Data are presented as means ± SD. Source data and exact *P* values are provided in the Source Data file.

($4.77 \pm 1.32$ mm$^3$) and the MS groups ($25.63 \pm 4.15$ mm$^3$, $P < 0.001$, Fig. 5b). Sagittal and coronary sections further demonstrated that the infectious calvarial bone defects almost healed in the sp-EMS group, while the defect center was discontinuous in the MS group, and little new bone was formed in the blank group. HE and Masson's trichrome staining showed a microstructure similar to that of the engineered tissues in the μCT images (Fig. 5c). The defect area was almost filled with new bone structures in the sp-EMS group, and no obvious remnant scaffolds were found. In contrast, a large amount of remnant scaffold was observed in the MS group. These results are consistent with in vitro experiments, indicating the great potential of the sp-EMS in infected bone repair.

To better understand the effects of the sp-EMS on the inflammatory responses and macrophage activation during infected bone regeneration, we further performed immunofluorescence staining in defect areas after 2 weeks of implantation (Fig. 5d–f; Supplementary Fig. 14). The percentage of TNF-α$^+$ cells in the sp-EMS group ($3.87 \pm 3.02\%$) was much lower than that in the MS ($48.22\% \pm 12.21\%$) and blank groups ($58.69 \pm 4.05\%$, $P < 0.001$). The same tendency was detected in the percentage of iNOS$^+$ cells. In addition, much more CD68$^+$CD163$^+$ M2 macrophages ($64.81 \pm 6.12\%$) were observed in the sp-EMS group than those in the MS ($10.74 \pm 0.53\%$) and blank groups ($6.64 \pm 0.49\%$, $P < 0.001$) after 2 weeks of implantation, whereas the number of CD68$^+$iNOS$^+$ M1 macrophages ($5.51\% \pm 2.26\%$) was much lower than that in the MS ($41.16 \pm 8.60\%$) and blank groups ($53.49\% \pm 2.94\%$, $P < 0.001$). These results indicated that implantation of the sp-EMS into the infected calvarial defects provokes M2 macrophage polarization, which alleviates inflammation status and promotes the bacteria-infected bone regeneration.

## The sp-EMS regenerates in situ rabbit open bone defects and dog vertical bone defects in a complex bacterial microenvironment

To investigate the potential of the sp-EMS to promote infected bone regeneration in a complex bacterial microenvironment, an open bone defect model in rabbits was created in the oral cavity with a larger number of opportunistic pathogenic species. The mandibular first premolar was extracted first, and then the cell-free sp-EMS and MS were implanted into the extraction socket without suture. Open bone defects without implantation of scaffolds were used as a negative control (Fig. 6a). After implantation for 8 weeks, reconstructed μCT images and corresponding quantitative results showed that ~98.38% of bone defects in the sp-EMS group were filled with native-like fibrous bone structures, including the cortical bone and the cancellous bone (Fig. 6b, Supplementary Fig. 15). In contrast, less new bone with discontinuous cortical bone structures were formed in the MS group with bone volume/total volume (BV/TV) of ~76.90% and in the blank group with BV/TV of ~66.65%. Notably, the bone mineral density of the engineered neobone formed by the sp-EMS ($0.49 \pm 0.06$ g/cm$^3$) was similar to that of natural trabecular bone ($0.54 \pm 0.02$ g/cm$^3$), which was much higher than that in the MS group ($0.28 \pm 0.01$ g/cm$^3$) and blank group ($0.16 \pm 0.01$ g/cm$^3$, $P < 0.001$). Similarly, the average trabecular thickness of newly formed alveolar bone in the sp-EMS group ($0.27 \pm 0.02$ mm) was significantly greater than that in the MS group ($0.19 \pm 0.01$ mm, $P < 0.001$) and the blank group ($0.11 \pm 0.01$ mm, $P < 0.001$). Thus, the sp-EMS could resist the complex bacterial microenvironment in the oral cavity and promote nearly complete in situ infected bone regeneration (Fig. 6b).

Next, histological analysis including HE and Masson's trichrome staining, was performed to observe the microstructure in the defect area. Consistent with the μCT results, large amounts of newly formed bones with blood vessels filled the defect area in the sp-EMS group, and no remnant scaffolds were observed. Masson's trichrome staining showed that the pattern of collagen fibers newly formed by the sp-EMS was very similar to the normal architecture of natural bones. In

contrast, in the MS group, limited new bone structures with a large number of undegraded scaffold remnants were identified in the defect center, and the cortical bone along the defect margin was discontinuous. In addition, in the blank group, only a small amount of new bone was formed in the defect center, and almost no new cortical bone formed along the defect margin (Fig. 6b). These results indicate that rabbit alveolar bone defects after tooth extraction within a complex bacterial microenvironment could be repaired by sp-EMS implantation.

The above data demonstrate that the sp-EMS exhibits multiple functions of osteo- and angioinduction and bacterial inhibition, which promote infected bone regeneration in rat calvarial and rabbit alveolar bone defect models by generating electrostimulation. To facilitate the clinical translation of the sp-EMS, critical-sized vertical full-thickness bone defects were created in the mandibles of beagle dogs whose jaws are very physiologically and pathologically similar to humans[31]. Initially, teeth were extracted, and the sockets were allowed to repair for 3 months. After that, alveolar bone atrophied, and $10 \times 10 \times 8$ mm$^3$ critical-sized full-thickness bone defects were created in oral bacterial microenvironment. Then, the cell-free scaffolds including the sp-EMS, MS, and Bio-Oss® collagen as a positive control were transplanted into the defect areas. Bone defects without implantation of scaffolds were used as negative controls (Fig. 6c; Supplementary Fig. 16). At 12 weeks postimplantation, μCT images and corresponding quantitative results showed that in the sp-EMS group, the defects were almost filled with native-like alveolar bone structures with BV/TV of $85.20 \pm 4.23\%$. In contrast, less new bone with discontinuous bone structures was formed in the MS group and Bio-Oss group, with BV/TV values of $62.55 \pm 3.76\%$ and $54.80 \pm 2.17\%$, respectively. Little new bone was formed in the blank group, with BV/TV of $31.69 \pm 4.36\%$. Moreover, the vertical bone height was almost completely repaired by the sp-EMS ($6.42 \pm 0.16$ mm), which was much greater than that in the blank group ($2.79 \pm 0.23$ mm, $P < 0.001$), Bio-Oss group ($4.70 \pm 0.34$ mm, $P < 0.01$) and MS group ($5.01 \pm 0.63$ mm, $P < 0.01$). In addition, the bone mineral density of the engineered neobone formed by the sp-EMS ($0.35 \pm 0.04$ g/cm$^3$) was much higher than that in the MS group ($0.26 \pm 0.03$ g/cm$^3$, $P < 0.01$), Bio-Oss group ($0.19 \pm 0.01$ g/cm$^3$, $P < 0.001$) and blank group ($0.04 \pm 0.02$ g/cm$^3$, $P < 0.001$) (Fig. 6d).

In addition, consistent with the μCT results, HE and Masson's trichrome staining showed that in the sp-EMS group, a large amount of newly formed bone was observed, and the height of the vertical defects was almost repaired with plenty of blood vessels filling the defect area. The pattern of newly formed collagen fibers in the sp-EMS group was very similar to the normal architecture of natural bones. In contrast, limited new bone structures were observed in the defect areas in the MS group and the Bio-Oss group, and the height of the vertical defects was not repaired. In addition, a number of undegraded scaffold remnants could be seen in the Bio-Oss group; whereas in the blank group, the vertical height of the defect area was barely repaired (Fig. 6e). Together, these data suggest that the sp-EMS could significantly induce vertical ingrowth of blood vessels and facilitate in situ bone formation in beagle dog full-thickness bone defects in oral bacterial microenvironment.

## Discussion

Bacterial infection is the main cause of the failure of bone regeneration since traditional bone substitutes lack self-antibacterial properties[32,33]. Currently, bone graft materials combined with antibiotics are commonly utilized to treat infected bone defects. However, antibiotics not only have difficulty dealing with complex infections but also inhibit the osteoinductive properties of bone grafts[34–36]. The native bone environment possesses an endogenous electroactive interface that induces stem cell differentiation and inhibits bacterial adhesion and activity[12,29]. Mimicking a native bioelectrical environment, we developed the sp-EMS to improve the antibacterial and regenerative potentials of bone grafts.

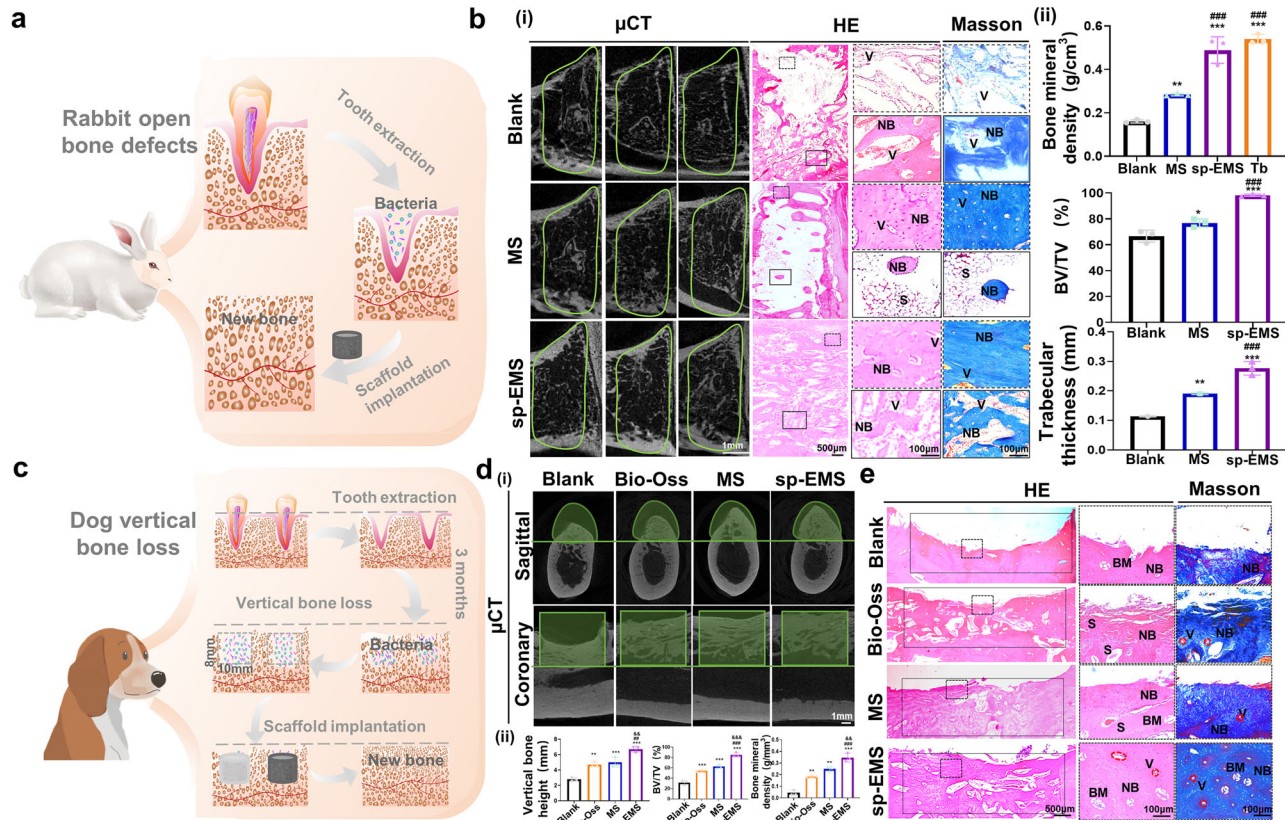

**Fig. 6 | The sp-EMS regenerates in situ rabbit open bone defects and dog vertical bone defects in a complex bacterial microenvironment. a** Schematic of in situ infected bone regeneration of rabbit open alveolar bone defects by the cell-free sp-EMS. **b** (i) Representative µCT, HE, and Masson's trichrome staining of the rabbit engineered bones in the blank, MS, and sp-EMS groups after 8-week implantation. Green lines display the defect boundaries. Black boxed areas and dotted box areas show high magnification views. (ii) Semiquantification of the bone mineral density, BV/TV, and the trabecular thickness in different groups in (i) (*n* = 3 rabbit open bone defects per group, by one-way ANOVA with Tukey's post hoc test: *$P < 0.05$, **$P < 0.01$, ***$P < 0.001$ versus blank; ###$P < 0.001$ versus MS). S scaffolds, NB new bone, V vessels, Tb trabecular bone. **c** Schematic of in situ vertical bone regeneration in beagle dogs by the cell-free sp-EMS. **d** (i) Representative µCT images of vertical bone defects in beagle dogs in the blank, Bio-Oss, MS, and

sp-EMS groups after 12-week implantation. Green boxed areas display the defect areas. (ii) Semiquantification of the vertical bone height, BV/TV, and bone mineral density in the blank, Bio-Oss, MS and sp-EMS groups in (i) (*n* = 3 beagle dog vertical bone defects per group, by one-way ANOVA with Tukey's post hoc test: **$P < 0.01$, *** $P < 0.001$ versus blank; ##$P < 0.01$, ###$P < 0.001$ versus Bio-Oss; &&$P < 0.01$, &&&$P < 0.001$ versus MS). **e** Representative HE and Masson's trichrome staining of the engineered bones in the blank, Bio-Oss, MS, and sp-EMS groups after 12-week implantation. NB new bone, V vessel (red circles), BM bone marrow (white circles), S scaffolds. Black box areas show the defect areas. Dotted box areas show high-magnification views. All experiments were repeated independently at least three times. Data are presented as means ± SD. Source data and exact *P* values are provided in the Source Data file.

Endogenous electrical properties-mediated stem cell differentiation/trans-differentiation into specific cell lineages is of great significance for bone regeneration[14,37]. Conventional electrical stimulation strategies rely on piezoelectric-based electroactive materials or wearable/implantable electrostimulation devices[14]. Piezoelectric materials have been shown to regulate the behavior and fate of stem cells by altering cell membrane potential or activating ion channels through the generated piezoelectric potential[38]. However, the application of piezoelectric materials in tissue engineering encounters some challenges. First, the heterogeneous interface of traditional piezoelectric materials, including piezoelectric polymers and inorganic crystals, are quite different from that of natural bone. During bone growth, the unique chemical composition and hierarchical micro-nanotopology of natural bone have been shown to play crucial roles in the recruitment and differentiation of stem cells. Although piezoelectric materials can activate ion channels through electrical stimulation, their osteoinductive capacity is hardly comparable to that of natural bone interface due to the difficulty in simulating the microenvironment of bone in terms of chemical composition and surface topology[39]. Second, most piezoelectric materials are highly dependent on the excitation of external energy fields including ultrasonic waves, to perform electrical

stimulation, and this reliance on exogenous energy excitation is not conducive to continuous, sequential, uninterrupted treatment. Although it has been reported that some piezoelectric films can generate electrical signals through deformation caused by traction force generated by cells attached to the surface, the accompanying electrical signals tend to be weak due to the small traction force of the cells. In addition, cellular force-driven piezoelectricity is also highly dependent on cell state and susceptible to interference from non-specific adsorption of biomacromolecules[40,41]. Therefore, implementing long-acting and effective electrical stimulation is also a huge challenge. Third, strong piezoelectric effect may inhibit cell activity and is not conducive to the osteogenic differentiation of stem cells[42,43]. Finally, to meet the requirements of bone regeneration in vivo, the biosafety and degradability of piezoelectric biomaterials also need to be carefully examined. However, achieving a balance between degradation and bone regeneration remains a challenge for conventional piezoelectric biomaterials[14]. For wearable/implantable devices, electrical stimulation can be implemented by connecting power sources, energy storage devices, and radio frequency induction systems[44]; however, difficulties also exist in tissue engineering applications. As a primary concern, both wearable and implantable electrical stimulation medical devices

inevitably embed some circuits (e.g., wires or electrodes) in the body, which poses biotoxicity risks and biosafety issues. In addition, wearable/implantable devices are also limited by the accessibility of power supply or charging equipment, making them more conducive to the implementation of intermittent electrical stimulation than to long-term treatment[45,46]. Finally, ideal tissue engineering materials require a rate of biodegradation commensurate with the rate of neotissue formation; however, for electrical stimulation devices, even transient electronic devices have difficulty meeting this requirement, posing the risk of secondary surgery[47]. Therefore, the development of highly biocompatible scaffold materials that can perform sustained electrical stimulation without relying on external energy input is urgently required. Inspired by the nanoscale galvanic cells[22], we developed the circuit-free sp-EMS with an electrical stimulation mechanism derived from the electrochemical corrosion reaction of silver nanostructures driven by the coassembled interface of Ag uNWs and mineralized collagen fibrils. The following points are worth mentioning: (i) The flexibility and spatial configurational freedom of ultrathin silver nanowires allowed them to be coassembled inside the sp-EMS fibers, instead of being exposed to the surface. The exposed surface of the sp-EMS was highly biocompatible mineralized collagen, similar to the chemical composition and micro-nanotopology of natural bone. The bone-like microenvironment of the sp-EMS was favorable for stem cell recruitment and osteogenic differentiation. (ii) The sp-EMS could self-promote the sustained release of weak electric current through mild electrochemical reaction and accompanying ionic displacements. Therefore, the self-promoted electrical stimulation based on the sp-EMS does not rely on external excitation sources, and will not be interfered by the cell adhesion state and non-specific adsorption of biomacromolecules. (iii) The sp-EMS produced mild electrical stimulation, which could promote the proliferation and differentiation of stem cells. (iv) The sp-EMS had suitable biodegradability. The animal experiments showed that the sp-EMS gradually degraded after 2 weeks of self-promoted electrical stimulation, and completely degraded at 12 weeks when bone defects were healed. These findings indicate that the sp-EMS could achieve a good balance between material degradation and bone regeneration. Moreover, electrochemical products ($Ag^+$) inhibited bacterial colonization and multiplication under electrical stimulation. The sp-EMS possessed excellent osteoinductive and degradation properties, and its metabolites could be removed from the body after repairing infected bone defects.

Electrical stimulation modulates bone formation and maturation by activating of voltage-gated calcium channels, followed by the amplification of cytosolic calcium before boosting intracellular calcium storage[24]. In the present study, an increasing $Ca^{2+}$ flow in BMSCs cultured on the sp-EMS was observed. In addition to regulating Ca ions, it has been proved that direct current stimulation can guide the migration of protons to reach mitochondrial membrane-bound H1-ATPases, thereby promoting high-level synthesis of ATP[16,48]. Furthermore, self-promoted electrical stimulation could also produce a similar phenomenon, where purinoceptor (P2X7) was significantly upregulated in the BMSCs cultured on the sp-EMS, indicating of higher ATP release. Intracellular ATP is consumed for the conversion of monomeric G-actin to polymeric F-actin, which promotes the reorganization of the actin cytoskeleton in BMSCs[48,49], ultimately altering the mechanical continuum between mechanosensitive cytoskeleton and nuclear scaffolds, and enabling regulation of downstream molecular signaling and gene expression[50]. Our work demonstrated that both F-actin levels and F-actin/G-actin ratios were significantly increased in BMSCs incubated on the sp-EMS, which confirms the reorganization of cytoskeleton after the continuous electrical stimulation from the sp-EMS, and ultimately enhances osteogenic differentiation of BMSCs through activation of the BMP2/Smad5 pathway. In previous studies, Ag nanostructures were considered to have cytotoxicity concerns, which were also related to their shape, size, and geometry of the

nanostructures[51]. However, in our study, sp-EMS exhibited excellent biocompatibility, which was attributed to the flexibility and spatial configurational freedom of Ag uNWs allowing them to coassembled inside the sp-EMS fibers, instead of being exposed to the surface and directly contacting the cells, thus constituting a highly biocompatible interface. To further confirm that the osteogenic effect of sp-EMS was derived from self-promoted electrical stimulation rather than the chemical composition of silver, we designed passivation structures (AgCl and $Ag_2S$ NWs) based on Ag uNWs to shield the electrochemical reactions of the scaffolds. Blocking electrical signaling significantly impaired the osteogenesis of BMSCs, confirming that sp-EMS regulated BMSC functions mainly through self-promoted electrical stimulation rather than Ag nanostructures.

Infected bone regeneration requires a highly coordinated repair response involving osteogenesis and antibacterial properties[52]. Conventional treatments try to incorporate antibacterial components into scaffolds[53]. However, the antibiotic concentration required for local slow release is low, which makes it easy for bacteria to resist, resulting in poor efficacy of infected bone repair[6]. In our study, sp-EMS achieved an excellent antibacterial effect against *S. aureus* by releasing the electrochemical intermediate product $Ag^+$ and ROS generated by self-promoted electrical stimulation. In the subsequent animal experiments, we established rabbit open bone defects and critical-sized vertical bone defects in beagle dogs in a complex bacterial microenvironment, which are clinically difficult[54], to verify the clinical translation potential of the sp-EMS. As expected, due to the perfect balance of osteogenic effects and antimicrobial properties, the sp-EMS achieved complete or nearly complete in situ bone regeneration.

Taken together, the sp-EMS provides a bone-like electrical environment, which promotes stem cell osteogenic differentiation and inhibits bacterial activity, and ultimately induces in situ infected bone regeneration. This study develops a passive electroactive degradable tissue engineering scaffold independent of external field excitation, which has great potential for clinical translation in the treatment of complex defect tissues.

## Methods

### Synthesis of silver ultrathin nanowires

Ag uNWs were synthesized via thermal decomposition of silver nitrate precursor in ethylene glycol solvent and reduction by polyvinylpyrrolidone (PVP)[55]. First, PVP (50 mg, MW = 1,300,000, Aladdin) and $AgNO_3$ (45 mg, Aladdin) were mixed in ethylene glycol (9.7 mL). After the solids were dissolved, NaBr (23 mg/mL in ethylene glycol, 0.2 mL, Aladdin) was added to the solution. The mixture was stirred (1200 rpm) at room temperature for 30 min until it turned opaque white. Then, benzoin (500 mg, Aladdin) was added and mixed thoroughly. After 7 min of bubbling $N_2$ through the reaction mixture, the mixture was heated to 160 °C for approximately 15 min ($N_2$ bubbling was maintained). Thereafter, $N_2$ bubbling was stopped, and the mixture was left undisturbed for 60 min. After cooling to room temperature, acetone was slowly added to the mixture to purify by selective precipitation.

### Fabrication of MS and sp-EMS

The MS was fabricated using our previously developed biomimetic self-assembly approach[19,20]. Briefly, type I tropocollagen solution (3.88 mg/mL, Corning®) was reconstituted in a dialysis flask (3500 Da, Solarbio) by immersion in a potassium buffer containing 200 mM KCl, 30 mM $Na_2HPO_4$, and 10 mM $KH_2PO_4$ (Aladdin) at 30 °C for 2–3 days. For the synthesis of sp-EMS, Ag uNW solution (10 mg/mL) was first mixed with type I collagen (3.88 mg/mL) at a ratio of 1:19 (v/v) and then the mixture was further placed in the potassium buffer. The fibrillized collagen was collected by centrifugation and stirred until a just-castable suspension formed. Then, the suspension was poured into the

mold, frozen for 24 h at −20 °C and lyophilized to create 3-D sponge-like porous scaffolds. To fabricate the MS and sp-EMS on culture plates, 1 mL of 1 mg/mL type I tropocollagen solution or the mixture (collagen: Ag uNW = 1:19) was dripped onto the bottom of the well and left to gel by incubation at 30 °C for 2–3 days. The bottom of the culture plate was observed by SEM to confirm that the material covered the entire plate. Finally, the assembled collagen was cross linked and mineralized in a solution containing 9 mM $CaCl_2$, 4.2 mM $K_2HPO_4$ and 0.2 mg/mL polyaspartic acid (Aladdin) at room temperature for 5–7 days (Supplementary Table 1).

## Synthesis of passivated silver nanowires

Passivated silver nanowires (Ag@AgCl uNWs, Ag@Ag$_2$S uNWs) were synthesized by a solution-phase reaction. To build up the passivation layer on the Ag uNW surfaces, sodium chloride and sodium sulfide were used as chlorine and sulfur sources, respectively. For the synthesis of Ag@AgCl uNWs, 20 mg PVP was added to 5 mL of the Ag uNW water solution (10 mg/mL) and then sonicated for 10 min. Subsequently, 5 mL sodium chloride (Aladdin) water solution (2 mg/mL) was slowly added to the mixture and stirred for 1 h at room temperature. The product was collected by centrifugation at the speed of $2800 \times g$ and washed with water and ethanol three times. To synthesize Ag@Ag$_2$S uNWs, 5 mL sodium sulfide (Aladdin) water solution (2 mg/mL) was added to 5 mL of the Ag uNWs water solution and stirred for 1 h at room temperature. The product was collected by centrifugation at the speed of $2800 \times g$ and washed with water and ethanol three times. All of the above products were dispersed into deionized water for subsequent characterization and testing.

## Transmission electron microscopy

The nanostructure of the sp-EMS fibrils was observed by TEM (JEM-100CX) at 100 kV. Samples were deposited onto Formvar carbon-coated nickel grids for analysis. For cells and bacteria, samples were fixed with 2.5% glutaraldehyde (Solarbio) and embedded in epoxy resin. Sections of 70–100 nm thickness were cut with a Leica ultramicrotome, collected on Formvar carbon-coated nickel grids, and then stained with 1% sodium phosphotungstate prior to TEM.

## ICP-OES instrumentation

The elemental analyses were carried out with the ICP-OES device (Thermo Fisher Scientific). The ICP-OES operating conditions were 0.8 L min$^{-1}$ nebulizer flow rate, 14 L min$^{-1}$ coolant flow rate, and 0.9 L min$^{-1}$ auxiliary flow rate.

## Scanning electron microscopy

The surface microstructure of the sp-EMS was observed by SEM (Hitachi S-4800) at 15 kV. After dehydration in a series of graded ethanol (50–100%), all samples were critical-point dried and sputter-coated with gold for 3 min at 20 mA.

## Atomic force microscopy

The nanomechanical properties of the sp-EMS and Ag uNWs were characterized by AFM (Dimension Icon, Bruker, USA). Under ambient conditions (room temperature), the sp-EMS and Ag uNWs were separately assembled onto freshly cleaved mica, and at least three scanning areas were imaged for each sample. The obtained $512 \times 512$ pixels property maps were analyzed using Nanoscope software.

## X-ray diffraction

XRD was carried out by using a D8 Advance (Bruker, Germany; at 45 kV and 40 mA) to characterize the crystal structures in different scaffolds. To obtain fine powders, all samples were thoroughly rinsed with deionized water, dried at the critical point and pulverized in liquid nitrogen. JADE 6.5 software was used to analyze the XRD patterns.

## Thermogravimetric analysis

The mineral content in scaffolds was determined by TGA (Mettler Toledo) at a heating rate of 10 °C/min in nitrogen over a temperature range from room temperature to 800 °C. The lyophilized sp-EMS and MS were cut into small pieces, and 3 mg of each sample was used for the TGA measurement.

## X-ray photoelectron spectroscopy

The elemental composition of the sp-EMS and MS was tested by XPS (Escalab 250Xi, Thermo Fisher, USA). The lyophilized scaffolds were cut into small pieces for measurement.

## Electrochemical measurements

Electrochemical tests were performed using a typical three-electrode configuration on CHE 760E (Shanghai Chenhua Instrument Co., China) instruments. PBS (pH = 7.2–7.4) was used as an electrolyte. The electroactive scaffold-loaded ITO glass electrodes, pure ITO glass electrodes, and Ag/AgCl electrodes (3.0 M KCl) were used as the working, counter, and reference electrodes, respectively. All samples including MS, sp-EMS, MS/AgCl, and MS/Ag$_2$S were coated directly on the $1.5 \times 2$ cm$^2$ ITO glass (Zhongjingkeyi Technology Co., Ltd, China) served as the working electrodes. The obtained electrode was mineralized at room temperature for 5–7 days, and the film was obtained for further measurements. Details of the preparation of the working electrodes are as follows: For MS-loaded ITO, 150 μL collagen, 150 μL acetic acid, and 300 μL potassium buffer were uniformly coassembled on the ITO glass as MS. For sp-EMS-loaded ITO, an aqueous solution of Ag uNWs (10 mg/mL, 25 μL) was added to the above collagen mixture and coassembled on ITO glass. The electrochemical test samples of MS@AgCl and MS@Ag$_2$S were prepared in the same way. Finally, electrocatalyst-loaded ITO glass was put into a vacuum drying oven and mineralized for five days for subsequent electrochemical measurements.

Before recording the electroactivity, several fast cyclic voltammograms (CVs, 500 mV s$^{-1}$) were taken to clean and stabilize the electrocatalyst surface. The Ag/AgCl (3 M KCl) electrode was calibrated with respect to the reversible hydrogen electrode (RHE) via the Nernst equation (Eq. 1). The CV curves were recorded between 0 and 1.0 V vs. RHE at a scan rate of 5 mV s$^{-1}$ in PBS (pH = 7.2–7.4).

$$E_{RHE} = E_{Ag/AgCl} + 0.0591pH + E^0_{Ag/AgCl} \qquad (1)$$

where $E_{RHE}$ is the converted potential vs. RHE, $E_{Ag/AgCl}$ is the experimentally measured potential against the Ag/AgCl reference electrode, and $E^0_{Ag/AgCl}$ is the standard potential of Ag/AgCl at 25 °C (0.197 V). The electric current test was recorded by taking a chronoamperometric curve at open circuit voltage. To eliminate unnecessary errors, the above tests were performed using a typical two-electrode system, and the counter electrode was ITO with the same area ($1.5 \times 2$ cm$^2$). All data are presented without iR compensation, and all electrochemical tests were performed at room temperature.

## Cell culture and differentiation

Human BMSCs (PCS-500-012) were purchased from Beijing Zhongyuan Company Limited (Beijing, China). and cultured in complete medium (alpha modification of Eagle's medium/10% fetal bovine serum/100 U/mL penicillin/100 μg/mL streptomycin) in a humidified atmosphere of 5% CO$_2$ at 37 °C. For the following studies, cells at passages 3–5 were seeded on the 6-well plate, MS, and sp-EMS, at a density of $2 \times 10^5$ cells per well. The scaffolds were sterilized under ultraviolet light for 2 h before cell seeding. The cells were cultured with the medium refreshed every two days and observed by a Zeiss light microscope to ensure the attachment of cells to the scaffold.

## Cell counting Kit-8 assay

The biosafety and cell proliferation of the sp-EMS was determined by the CCK-8 assay (Solarbio, Cat# 1210). BMSCs were seeded on the 6-well plate (blank), MS, and sp-EMS on days 1, 3, and 7. The samples, at desired time points, were incubated in CCK-8 solution at 37 °C for 2 h. The absorbance was read at 450 nm using a microplate reader (Bio-Rad, Japan).

## Measurement of intracellular calcium levels

BMSCs seeded on the 6-well plate, MS, and sp-EMS for 7 days were used to measure intracellular calcium. Intracellular calcium levels in cells were measured with Fluo-3 calcium assay kits (Fluo-3 AM, Solarbio, Cat#121714-22-5) according to the manufacturer's instructions. Briefly, after removing the medium, 100 μL Fluo-3 was added to each sample and incubated at 37 °C for 60 min. After washing with PBS, the cells were mounted with mounting media containing DAPI (ZSGB-BIO, Cat#ZLI-9557) and observed by a Leica SP8 laser scanning microscope.

## Live/dead staining

Cells were seeded on 35-mm confocal dishes precoated with scaffolds at a density of ~1 × 10⁵ cells per well. Then, the cells were stained with a calcein AM/PI double-stain kit (Solarbio, Cat#1630) according to the instructions of the manufacturer for CLSM imaging. All the samples were observed by a Leica SP8 laser scanning microscope.

## Quantitative real-time polymerase chain reaction

To investigate the gene expression of human BMSCs in the 6-well plate (MS and sp-EMS groups), cells were cultured for 7 and 14 days in osteogenic differentiation conditioned medium (regular medium supplemented with $10^{-7}$ M dexamethasone, $10 \times 10^{-3}$ M β-glycerophosphate and $0.05 \times 10^{-3}$ M ascorbic acid 2-phosphate). In addition, cells in osteogenic differentiation conditioned with 10 μg/mL *S. aureus*-derived lipoteichoic acid (LTA) (Sigma-Aldrich, Cat# L2515) stimulation were cultured to investigate inflammatory-related gene expression. Quantitative RT-PCR was applied to examine the expression of osteogenic differentiation gene markers, including bone morphogenetic protein-2 (*BMP2*) and osteocalcin (*OCN*), calcium channel L type DHPR alpha 2 subunit (*CACNA2D1*) and purinoceptors (*P2X7*). Glyceraldehyde 3-phosphate dehydrogenase (*GAPDH*) was used as the housekeeping gene. To demonstrate the accuracy and stability of *GAPDH* as the housekeeping gene in interpretation of relative gene expression after electrical stimulation for 24 h, additional endogenous controls *PPIA* and *YWHAZ* were tested[56]. The expression levels of these two genes relative to *GAPDH* are similar (Supplementary Fig. 17).

Total RNA was extracted by TRIzol reagent (Thermo Fisher Scientific, Cat#15596026), and synthesis of cDNA was performed using the SuperScript III One-Step RTPCR System with Platinum Taq High Fidelity (Invitrogen). Quantitative RT–PCR was performed on a 7900HT Fast real-time PCR machine (Applied Biosystems) using SYBR Green (Invitrogen Life Technologies). The primers were designed by Primer Premier 5.0 software and commercially synthesized (Supplementary Table 2).

## Western blotting

The cell lysate proteins were harvested by RIPA buffer (Thermo Fisher Scientific, Cat#89900) containing Protease/Phosphatase Inhibitor Cocktail (Thermo Fisher Scientific, Cat#87786) on ice. The lysates were centrifuged at 12,000 × *g* for 20 min at 4 °C. Protein quantification was performed using a Pierce BCA protein assay kit (Thermo Fisher Scientific, Cat#23225). The absorbance at 595 nm was measured using a microplate reader (Bio-Rad). Then, equal amounts of proteins (20–30 μg) were mixed with 5× loading buffer (Solarbio, P1040) and boiled at 99 °C for 10 min. The samples were separated by 10% SDS−polyacrylamide gel electrophoresis, transferred to polyvinylidene difluoride membranes and blocked in 5% skim milk in 0.1% TBS-Tween (Sigma-Aldrich, P9416). The membranes were separately probed with corresponding primary antibodies against GAPDH (1:1000, Proteintech, Cat#60004-1-Ig), BMP2 (1:1000, Abcam, Cat#AB214821), F-actin (1:1000, Cytoskeleton, Cat#BK037), Actin (1:1000, ZSGB-BIO, Cat#TA-09), p-Smad1/5/9 (1:1000, CST, Cat#13820), and Smad5 (1:1000, CST, Cat#12534), at a dilution of 1:1000 in 5% w/v skim milk overnight at 4 °C. Membranes were washed three times in TBS with 0.1% Tween-20, incubated with an HRP-conjugated secondary antibody (1:10000, diluted with TBST, ZSGB-BIO, Cat#ZB-2301 and Cat#ZB-2305) for 1 h, and imaged by an Odyssey® Imaging System (Supplementary Table 3).

## F-actin/G-actin ratio assay

To analyze the F-actin/G-actin ratio, homogenates were fractionated by differential ultracentrifugation and the relative levels of filamentous and globular actin in each fraction were quantified. According to the manufacturer's instructions (Cytoskeleton, BK037, Denver, CO.), each sample was placed into a 25 G syringe and quickly homogenized in 200 μL of lysis buffer at 37 °C. Then, the cell samples were placed on ice until all samples were completed. The homogenates were transferred to a prewarmed (37 °C) ultracentrifuge and spun at 150,000 × *g* for 1 h at 37 °C to separate the G-actin (supernatant) and F-actin fractions. The samples were stored at −80 °C until further separation by western blotting analysis.

## Electrical stimulation of human BMSCs

A signal generator was custom made to directly input a current of 4 μA to the BMSC medium through the electrode. The system consisted of three parts: a stimulation power supply (A-BF Programmable DC Power Supply, China), a platinum sheet for current conduction, and a cell culture device (Supplementary Fig. 7a). The cells were exposed to current stimulation for 48 h and their biochemical reactions were observed.

## Antimicrobial tests

*S. aureus*, the main pathogen of infected bone defects, was used as a typical gram-positive bacterium to assess the antibacterial ability of the sp-EMS. *S. aureus* CGMCC1.6722 was provided by the China General Microbiological Culture Collection Center (Beijing, China), and was cultured in Luria-Bertani (LB) medium (Sigma, Cat#L7275). CFU assay, live/dead staining, SEM, and TEM were used to assess the antibacterial properties of the sp-EMS. All scaffolds were precoated on 48-well plates and sterilized by ultraviolet light for 2 h. The bacteria were all suspended in the liquid LB medium. Two hundred microlitres of 1 × 10⁸ colony-forming units per milliliter (CFU/mL) bacterial suspension was added to each well. The 48-well plates were placed at 37 °C for 24 h.

For the CFU assay, after coculturing with the scaffolds for 24 h, 10 μL of bacterial suspension was diluted 1000 times with sterile LB liquid medium for further testing. According to the manufacturer's instructions, 50 μL of diluted bacterial suspension was spread on the solid LB agar plates by a Whitley Automated Spiral plater (Interscience, USA). After incubation at 37 °C for 24 h, the plates were removed to count the number of colonies.

For the live/dead staining, after coculturing with the scaffolds for 24 h, 100 μL of bacterial suspension was plated on 35-mm confocal dishes. Then, the bacteria were stained with a calcein AM/PI double-stain kit (Solarbio, Cat#1630) according to the instructions of the manufacturer for CLSM imaging. All the samples were observed by a Leica SP8 laser scanning microscope.

The micro-nano morphology of bacteria on different scaffolds was observed by SEM and TEM. Two hundred microlitres of 2.5% glutaraldehyde was added to the well to immobilize bacterial morphology for 2 h in the dark. After dehydration in a series of graded ethanol (30−100%), all samples were prepared for SEM and TEM observations. To quantify bacteria, three fields were randomly selected from each

slide and bacteria were counted. The average bacterial count across the three fields on each slide was determined, and the average of three slide counts was calculated for each sample.

## Ag⁺ release test

The sp-EMS were immersed in 10 mL of PBS at 37 °C. The liquid was exchanged every day from day 1 to day 14. Three milliliters of Ag⁺ release liquid was replaced with 3 mL PBS every other day. The amount of Ag⁺ release was examined by measuring the solution by ICP–OES (Thermo Fisher Scientific).

## Reactive oxygen species detection

The production of ROS is an important antibacterial property of the scaffolds. The amount of ROS was determined by 2',7'-dichloro-fluorescein diacetate (DCFH-DA) because it can capture ROS signals. ROS production was measured with an ROS assay kit (Solarbio, Cat#4091-99-0) according to the manufacturer's instructions. The whole process was carried out in a dark environment. 200 μL of self-assembled collagen fibrils were poured into 96-well plates and then lyophilized for use. After mineralization, all scaffolds were sterilized by ultraviolet for 2 h. Then, 100 μL of $1 \times 10^8$ colony-forming units per milliliter (CFU/mL) bacterial suspension was added to each well. After incubation for 6 h, 12 h, and 24 h, 10 μM DCFH-DA was added to each well for 20 min in the dark at 37 °C. The fluorescence intensity was tested using a multimode microplate detection system (Centro LB960, Berthold) (Ex = 488 nm, Em = 525 nm). ROS production was quantified by comparing the fluorescence values measured by the microplate reader to the blank control group, to obtain a relative ROS level.

## In vivo study design

The in vivo study design followed the principles of animal experiments from simple to complex, from small animals to large animals. To provide reliable data for the clinical transformation of the sp-EMS, we simulated different clinical application scenarios from rat calvarial defects with single bacterial infection, to rabbit open alveolar defects and beagle dog vertical bone defects in oral bacterial microenvironment. All animal experimental procedures used in the study were performed in compliance with animal welfare ethical regulations and approved by the Animal Use and Care Committee of Peking University (LA2022190). According to Canadian Council on Animal Care (CCAC) and American Veterinary Medical Association (AVMA) euthanasia guidelines, rats were euthanized by intraperitoneal injection of pentobarbital at 200 mg/kg body weight. Rabbits and beagle dogs were chemically euthanized using an overdose of intravenous pentobarbital sodium at 400 mg/kg body weight. All animals survived the surgery and recovery period. The details of the animals included in all the experiments were listed in the Supplementary Table 4.

In rat calvarial defects, we repeated multiple rounds of experiments to examine and confirm the therapeutic effect of sp-EMS. At each round of rat experiments, we included at least 5 rats in each group based on previously published priori power analyses[19,20,57]. Power analysis requires prior knowledge of two key parameters: (1) the effect size, defined as the minimum difference between groups considered clinically significant, and (2) the standard deviation, which measures variability within a sample for a quantitative variable. Without estimates of the effect size and standard deviation from previous similar studies, power analysis cannot be performed. Since this study is the first of its kind conducted in rabbits and dogs, it is impossible to assume the effect size and the standard deviation required for power analysis. For this reason, an alternative method, the 'resource equation' approach, which sets the acceptable range of degrees of freedom (DFs)[58], was used to determine the sample size. To achieve a DF between 10 and 20, the minimum number for each group was 2.67, whereas the maximum number for each group was 4.33[59]. Therefore, $n = 3$ was used for each group in the in vivo study of the rabbits and

beagle dogs, which follows the principles of the 3Rs (Reduction, Refinement, and Replacement) in the calculation of the sample size in animals. The time points chosen for animals to evaluate bone regeneration efficiency and mechanism were based on previously published works on mineralized scaffolds[19,20,60]. Specifically, the time points of 1–2 weeks were chosen to evaluate the inflammatory status in the defect area[61]; 8–12 weeks were used to assess bone recovery.

## A rat calvarial noninfected bone defect model

Male Sprague-Dawley rats (200–220 g, 6–8 weeks old) were used and maintained on a 12 h light/dark cycle in individually ventilated cages at 22 °C and 48% humidity with unrestricted access to food and water. During experiment procedures, rats were initially anesthetized by intraperitoneal injection of 1% pentobarbital sodium solution (Merck). After separating the skin and muscle, the calvarium was exposed. The whole layer of bone tissue was removed with a dental implanter (W&H Implantedmed, Austria) at 1200 rpm and critical-sized calvarial bone defects of 5 mm in diameter were created. Then different scaffolds were implanted into the defects and the skin was sutured with 6-0 surgical sutures. No perioperative antibiotic was used in these animals. Rats were randomly separated into three groups ($n = 5$): (i) blank group (untreated), (ii) MS group (implanted with the MS) and (iii) sp-EMS group (implanted with the sp-EMS). The postoperative pain was scored by grimace scale, a facial expression-dependent measure developed for quantifying spontaneous pain based on human facial coding scales[62]. When the pain score >2, the animal would be carefully examined. If the score > 4, additional intervention should be applied to the animals. After implantation for 2 weeks, rats from each group were sacrificed by overanesthesia via intraperitoneal injection of 200 mg/kg pentobarbital sodium to assess the Ca²⁺ influx level by quantifying the expression of CACNA2D1. After implantation for 8 and 12 weeks, rats were sacrificed by overanesthesia to evaluate bone regeneration efficiency.

To test whether self-promoted electrical stimulation accompanies the bone regeneration process, we measured the potential and current of the sp-EMS in situ at different periods after implantation using the electrometer (Keithley 6517B) and oscilloscope (Teledyne LeCroy HD 4096). The current was measured by placing the positive electrode of the electrometer in the scaffolds and the negative electrode at the calvarial bone away from the scaffolds for 5 min (Supplementary Movie 1).

## A rat calvarial infected bone defect model

After creating critical-sized calvarial bone defects with a diameter of 5 mm, 100 μL of $1 \times 10^8$ CFU/mL *S. aureus* suspension was applied to the defect areas and different scaffolds were implanted. Rats were randomly divided into three groups including the blank, MS, and sp-EMS groups ($n = 5$). After 2 weeks of implantation, rats from each group were sacrificed by overanesthesia to assess the inflammatory responses and macrophage activation by quantifying the expression of TNF-α, iNOS, CD68, and CD163. After 12 weeks of implantation, rats were sacrificed by overanesthesia, and the calvarial bones were obtained to evaluate the bone regeneration efficiency.

## An open rabbit alveolar bone defect model

To build an infected bone defect model in a complex bacterial microenvironment, five white adult female New Zealand rabbits (2.0–2.5 kg, 2.5–3 months old) were included (Supplementary Table 4). Rabbit bilateral mandibular first premolars were extracted after anesthetized by 2% pentobarbital sodium solution. Totally 10 alveolar bone defects were created, which was randomly divided into three groups including the blank, MS, and sp-EMS groups ($n = 3$). After implanting scaffolds into the extraction sockets, the sockets were completely open to the oral cavity. One broken tooth root was excluded due to accidental breakage. After 8 weeks of implantation, all rats were

sacrificed by overanesthesia with an overdose of intravenous pento-barbital sodium (400 mg/kg), and the mandibles were then detached and fixed in 4% paraformaldehyde.

### A critical-sized full-thickness alveolar vertical bone defects in beagle dogs

Three adult male beagle dogs (8–10 kg, 1–1.5 years old) were included and fed daily under a constant temperature environment (Supplementary Table 4). During surgery, body temperature of animals was maintained throughout the surgery using a homeothermic system. Vital signs of the animals were monitored daily after surgery. All pre-molars and first molars in the mandibles of the beagle dogs were extracted under anesthesia with 3% pentobarbital sodium solution. After healing for 3 months, two critical-sized full-thickness vertical bone defects of 10 mm × 10 mm × 8 mm$^3$ were created at each side of mandible and four defects were created in one dog. Totally 12 vertical bone defect areas were randomly divided into four groups ($n = 3$): (i) blank group (untreated), (ii) MS group (implanted with the MS), (iii) sp-EMS group (implanted with the sp-EMS), and (iv) Bio-Oss group (implanted with Bio-Oss® collagen as a positive control). After 12 weeks of implantation, all the beagle dogs were sacrificed by overanesthesia with an overdose 400 mg/kg of intravenous pentobarbital, and the mandibles were obtained and fixed with 4% paraformaldehyde.

### Microcomputed tomographic analysis

All fixed samples were scanned with a μCT system (Skyscan 1174, Bruker, Belgium) at 53 kV and 810 μA. The NRecon and CTvox software were used for three-dimensional image reconstruction. The region of interest of the entire bone defect area was selected semiautomatically. The bone volume, bone mineral density and thickness of the newly formed bone were measured by CTAn software (Supplementary Table 5).

### Histological assessment

After μCT analysis, all samples were demineralized in 15% ethylene-diaminetetraacetic acid for at least 4 weeks and then embedded in paraffin. Serial 5-μm-thick sections were prepared and dewaxed with xylene by exposure to a graded series of ethanol (70–100%). At least three sections from one individual sample were collected and stained with HE and Masson's trichrome and observed by a Zeiss light microscope.

### Immunofluorescent staining

The sections were subjected to antigen retrieval by using 0.125% trypsin and 20 μg/mL proteinase K at 37 °C for 30 min, blocked with 5% bovine serum albumin (BSA, Solarbio, Cat# A8010) for 1 h, and incubated with primary antibodies against CACNA2D1 (Abcam, Cat#AB238110), TNF-α (Abcam, Cat#AB183218), iNOS (Abcam, Cat#AB178945), CD68 (Proteintech, Cat#28058-1-AP), CD163 (Abcam, Cat#AB156769), at a dilution of 1:200 by 3% w/v BSA overnight at 4 °C. Next, FITC or Rhodamine-labeled secondary antibodies (ZSGB-BIO, diluted with PBS, Cat#ZF-0311 and Cat#ZF-0313) were applied to bind primary antibodies for 1 h. After washing with PBS, all samples were mounted with mounting media containing DAPI and viewed by a Leica SP8 laser scanning microscope or a Zeiss laser-scanning microscope. For semiquantification, the ratios of positive cells were calculated three times by a trained researcher who was blinded to the group design.

### Immunohistochemistry staining

The selected sections were deparaffinized, blocked and incubated with primary antibodies against BMP2 (Abcam, Cat#AB214821), OCN (Abcam, Cat#AB93876), VEGFR-1 (Abcam, Cat#AB2350), and CD31 (Abcam, Cat#AB32457) at a 1:200 dilution by 3% w/v BSA overnight at 4 °C. After washing, all samples were incubated with horseradish peroxidase-conjugated secondary antibodies (ZSGB-BIO, diluted with PBS, Cat#PV-9001) and observed under a Zeiss light microscope. Each group comprised at least 3 slides, and each slide was observed at the defect margin and the defect center including the scaffolds.

### Statistics and reproducibility

All experiments were repeated at least three times. The semi-quantitative analysis was obtained from at least three samples and presented as the means ± standard deviation (SD). Statistical differences were assessed by Student's t test (between two groups) and one-way ANOVA with Tukey's post hoc multiple comparisons test (among three or four groups) by using GraphPad Prism 8 software (GraphPad Software Inc.). A $P$ value less than 0.05 was considered statistically significant.

### Reporting summary

Further information on research design is available in the Nature Portfolio Reporting Summary linked to this article.

## Data availability

All data supporting the findings of this study are available within the article and its supplementary files. Any additional requests for information can be directed to, and will be fulfilled by, the corresponding authors. Source data are provided with this paper.

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

## Acknowledgements

This work was supported by the National Natural Science Foundations of China 82230030, 81871492 (Y.L.), 52372174 (D.L.), and 82170996 (D.H), Beijing International Science and Technology Cooperation Project Z221100002722003 (Y.L.), Peking University Clinical Medicine Plus X - Young Scholars Project PKU2023LCXQ004 (Y.L.),Ten-Thousand Talents Program QNBJ2019-2 (Y.L.), Key R & D Plan of Ningxia Hui Autonomous Region 2020BCG01001 (Y.L.), Innovative Research Team of High-level Local Universities in Shanghai SHSMU-ZLCX20212402 (Y.L.), National Key Research and Development Program of China 2022YFB3205600 (D.L.) and 2021YFB3201200 (D.L.).

## Author contributions

Y.L. and D.L. designed experiments, analyzed the data and prepared and revised the manuscript. Z.-X.L. and D.-Q.H. performed the experiments, analyzed the data, and performed the manuscript. B.-W. G., Z.-K.W. and M.Y. assisted with scaffold fabrication and characterization. H.-J.Y., Y.W., S.-S.J. and L.-S.Z. assisted with animal experiments. L.-Y.C., C.-Y.D., X.-L.W. and T.-H.W. performed some in vivo experiments. S.-Q. G. and J.M. analyzed the data. Y.-H. Z. revised the manuscript. All authors reviewed the manuscript.

## Competing interests

The authors declare no competing interests.

## Additional information

¹Laboratory of Biomimetic Nanomaterials, Department of Orthodontics, Peking University School and Hospital of Stomatology, Beijing 100081, PR China. ²Beijing Institute of Nanoenergy and Nanosystems, Chinese Academy of Sciences, Beijing 101400, PR China. ³Department of Stomatology, Peking University People's Hospital, Beijing 100044, PR China. ⁴National Center for Stomatology & National Clinical Research Center for Oral Diseases &National Engineering Research Center of Oral Biomaterials and Digital Medical Devices & Beijing Key Laboratory of Digital Stomatology & Research Center of Engineering and Technology for Computerized Dentistry Ministry of Health & NMPA Key Laboratory for Dental Materials & Translational Research Center for Orocraniofacial Stem Cells and Systemic Health, Beijing 100081, PR China. ⁵Fourth Clinical Division, Peking University School and Hospital of Stomatology, Beijing 100081, PR China. ⁶Center of Stomatology, Tongji Hospital, Tongji Medical College, Hubei Province Key Laboratory of Oral and Maxillofacial Development and Regeneration, Huazhong University of Science and Technology, Wuhan 430030, PR China. ⁷These authors contributed equally: Zixin Li, Danqing He. ✉e-mail: luodan@binn.cas.cn; orthoyan@bjmu.edu.cn

