## [Peer review file · Nature Communications]

REVIEWER COMMENTS

Reviewer #1 (Remarks to the Author):

The work reports on a electroactive mineralized scaffold that generates weak currents via spontaneous electrochemical reactions to activate voltage-gated Ca²⁺ channels, enhance ATP-induced actin remodeling, and ultimately achieve osteogenic differentiation of mesenchymal stem cells by activating the BMP-2/Smad5 pathway. Furthermore, the electroactive interface bacterial adhesion and activity via electrochemical products and concomitantly generated reactive oxygen species. Thus, a dual osteogenic and antibacterial dual function is achieved. Tests in vivo allow to validate the research hypothesis. ~

The work in is an area of increasing interest as scaffolds are increasingly being developed to simulate not just the static or dynamic morphological damaged tissue microenvironment, but also to provide the necessary biomimetic cues, either biochemical or biophysical.

In this case, electrical signals are convincingly demonstrated to provide advanced tissue regeneration and antimicrobial capabilities.

The work provides the complete description of the experimental work, allowing reproducibility, and the experimental data are properly discussed. The physical chemical analysis of the materials as well as the bio-related data are coherent and well discussed. Nevertheless, several issues must be provided before publication of the work:

-based on the piezoelectric nature of bone, piezoelectric scaffolds, mainly based on PVDF and co-polymers, as well as biodegradable piezoelectric polymers, have been proposed for bone tissue regeneration and it has been demonstrated that they also support antibacterial capabilities. The authors must present and discuss those results in comparison of the present ones, as they do not use nanoparticles, are based on a single material (and therefore are simpler) and they are not related to moving charges but with varying surface potential. The authors must discuss those issues to highlight novelty;

-the long-term potential issues related to the used of nanoparticles must be addressed (e.g. secondary reactions based on the large surface area and small dimensions, able to interact to cell components);

-Ag and Ag related reactions can have negative effects over time, once in the body. The authors must elaborate on that;

Reviewer #2 (Remarks to the Author):

- What are the noteworthy results?

1) Microcurrent of 4 μ A are generated spontaneously within the sp-EMS, which played a significant role in enhancing the bactericidal effects, angiogenic/osteoinductive properties of the scaffold.

2) Comparable results are documented in the in vitro experiments with stem cells and further validated in three different animal models.

- Will the work be of significance to the field and related fields? How does it compare to the established literature?

Yes the work will be of significance to the field bone tissue regeneration and development of electroactive bone graft materials. The synthesis of Ag nanowire, bactericidal effects/osteoinductive

property of Ag and the implementation in biomaterials for bone regeneration is not entirely novel. However, the determination of micro-current that is self-generated by electrochemical reaction of Ag nanowires occurring in the scaffold would be the unique aspect of this manuscript. Furthermore, the authors have also performed a rather all-rounded investigation; both in vitro and in vivo. Specifically, three different animal models are used and comparable results are documented. Their hypotheses are well supported with the presented results and have filled up some current gaps-in-knowledge in the field.

- If the work is not original, please provide relevant references.

Yes, the work is original.

- Does the work support the conclusions and claims, or is additional evidence needed?

The majority of work performed in this study has supported the conclusions and claims. The reviewer seeks the following items to further strengthen some conclusions:

1) The results clearly show the bactericidal effects and osteoinductive properties of sp-EMS (Fig 3, 4 and 5). Could the authors further commend the effects of sp-EMS on the inflammatory responses/macrophage activation at the site of critical defects in the animal models?

- Are there any flaws in the data analysis, interpretation and conclusions? Do these prohibit publication or require revision?

The data analysis and interpretation are nicely done and clearly presented. Some additional information listed below would certainly improve the manuscript.

1) Please include live measurements of electric current between the platinum sheets as Supplementary materials. So as to prove that the cells were indeed exposed to 4 μ A and there were no current depletion in the well.

- Is the methodology sound? Does the work meet the expected standards in your field?

The methodologies implemented in this study are sound and well-designed with accurate controls. They meet the expected standards and are sometimes above expectations. A job well done by the authors.

Some recommendations to further strengthen the presented results are:

1) The interpretation of relative gene expression with only GAPDH as the sole housekeeping gene could be less accurate. A recent study shows fluctuation in GAPDH after electrical stimulation and PPIA, YWHAZ are recommended (doi: <https://doi.org/10.3390/app12010153>). Nevertheless, additional endogenous control should at least be used with GAPDH and is recommended in future studies.

- Is there enough detail provided in the methods for the work to be reproduced?

Yes, sufficient details are provided by the authors.

- Additional areas for improvements and clarifications

1) The title is rather general the "Mineralized Scaffold" should be better defined. Perhaps "regeneration of bacteria infected bone defects" is a more appropriate term to use at the end.
2) "Infected bone loss" in line 25 is probably not a correct term, do the authors mean "Infected bone defect" or "Infection driven bone loss"?
3) Please check the references cited, as the citations do not explain the "Statements" made. For example:
a. Ref 1,2 (line 45): High treatment failure rate of infected bone defects are stated, but literatures for

drug-eluting joint implant and modifies antibacterial TiO₂ implants that showed improved treatment outcomes are cited.

b. Ref 9,10 (line 58-59): The term "Bioelectricity" is used. This encompasses dielectric, piezoelectric, pyroelectric, ferroelectric, streaming potential properties. Both citations are only referring to piezoelectric properties of tissues. Additionally "maintaining the physiological function of bone" is stated, but Ref10 mentioned only synthetic piezoelectric electrospun fibers. Do not see the link.

c. Ref 16 (line 70): self-polarization process of bone is stated in line 69, but Ref 16 is a study on modified titanium surface.

d. Ref 18 (line 101): It is stated that "Young's modulus of Ag uNW ranged from 0.22-0.59 GPa was similar to that of collagen fibrils". However, nowhere in the cited literature had listed these values. The lowest Young's modulus for EMC was 1.01 +/- 0.24 GPa. Also, are the authors referring to mineralized collagen fibrils? Please clarify.

4) The term "inflammatory tissues" (line 48) is perhaps misused? Is inflammatory response or inflamed tissues referred?

5) Antibiotics... "destroyed the body's microbial and immune environment,..."(line 54), do the authors mean antimicrobial environment?

6) "shear-force of collagen" line 60 refers to piezoelectric property of bone tissue upon application of mechanical force? The sentence is overly simplified, not sure what the authors mean here.

7) The term "almost dead" in line 357 seems inappropriate here, since there are no means of measurement to determine that.

Reviewer #3 (Remarks to the Author):

Rephrase this sentence, as it is not really clear and I do not know what latent reproduction is supposed to mean: The open environment of bone defects is conducive to the latent reproduction of various pathogenic bacteria.

This sentence should also be edited for clarity. Perhaps it may be split into two sentences, and it is not really correct to say antibiotics destroy the immune environment. Please adapt the sentence to have only correct statements: However, it is difficult for antibiotics to deal with complex infections caused by drug resistant bacteria; more seriously, antibiotics also inhibit the osteoinductive properties of bone grafts by destroying the body's microbial and immune environment, ultimately leading to a "double loss" of anti-infection and bone repair

I would like the introduction to provide more background information on the materials developed. What was characterised and tested prior to this paper?

Methods.

The DF approach for group size seems to suggest a low number of animals per group. This is not a very common approach. Does a group of 3 animals allow a robust evaluation of efficacy of the treatments?

In any case, I would like to see some data table listing all animal experiments, all groups per experiment and group size per experiment.

Since animal welfare is such an important issue, the authors are encouraged to follow best practice and disclose all relevant data pertaining to animal welfare.

The number of animals used per group per model should be disclosed. How was anaesthesia performed? How many animals were included but did not survive the surgery or recovery period? How was postoperative pain managed and scored? What were criteria for animal pain or burden to warrant intervention or early euthanasia? If the surgical procedures have not been described before, some details need to be given. For example, saying "Critical-sized calvarial bone defects of 5 mm in diameter were created and different 873 scaffolds were implanted" is not detailed enough to aid the reader who may want to perform this model. So add a reference for a full description, or provide more details here. Was any perioperative antibiotic prophylaxis done in these animals?

Details about how electrodes were applied to the animals, how long they were applied, and how the

animal tolerated the electrodes should be given.

Please provide more details for this process: Tissues in the defect area were obtained and placed into 5 mL of sterile LB agar medium. After co-cultured for 24 h, 50 μ L of bacterial suspension was coated on solid medium after diluted with PBS. Then, the culture dishes were placed in a bacterial incubator at 37 °C for 24 h and removed to examine the bacterial colony.

Why was the sample placed in LB medium? As it is written I have concerns that the method is not appropriate

For this text, is this 10 per animal? Totally 10 alveolar bone defects were 903 created, which was randomly divided into three groups.

The CT images of the bones may be better as 3D reconstructions instead of individual slices as this will show the entire bone.

The methods for the in vitro antimicrobial tests should be substantially clarified and improved: For CFU assay, the scaffolds and the bacteria suspension were co cultured for 24 h. Then 10 mL of sterile LB agar medium was poured into a petri dish for the next tests. According to the manufacturer's instructions, 50 μ L of bacterial suspension was coated on solid medium by a Whitley Automated Spiral plate.

As it is written I have concerns that the method is not appropriate

Responses to comments from the Reviewers

Reviewer #1

General Comment: The work reports on an electroactive mineralized scaffold that generates weak currents via spontaneous electrochemical reactions to activate voltage-gated Ca^{2+} channels, enhance ATP-induced actin remodeling, and ultimately achieve osteogenic differentiation of mesenchymal stem cells by activating the BMP-2/Smad5 pathway. Furthermore, the electroactive interface bacterial adhesion and activity via electrochemical products and concomitantly generated reactive oxygen species. Thus, a dual osteogenic and antibacterial dual function is achieved. Tests in vivo allow to validate the research hypothesis.

The work in is an area of increasing interest as scaffolds are increasingly being developed to simulate not just the static or dynamic morphological damaged tissue microenvironment, but also to provide the necessary biomimetic cues, either biochemical or biophysical.

In this case, electrical signals are convincingly demonstrated to provide advanced tissue regeneration and antimicrobial capabilities.

The work provides the complete description of the experimental work, allowing reproducibility, and the experimental data are properly discussed. The physical chemical analysis of the materials as well as the bio-related data are coherent and well discussed. Nevertheless, several issues must be provided before publication of the work.

Reply: We thank the Reviewer for these very positive comments. As suggested, we have made corresponding modifications as follows.

Q1: based on the piezoelectric nature of bone, piezoelectric scaffolds, mainly based on PVDF and co-polymers, as well as biodegradable piezoelectric polymers, have been proposed for bone tissue regeneration and it has been demonstrated that they also support antibacterial capabilities. The authors must present and discuss those results in comparison of the present ones, as they do not use nanoparticles, are based on a single material (and therefore are simpler) and they are not related to moving charges but with varying surface potential. The authors must discuss those issues to highlight novelty.

Reply: We appreciate the Reviewer's suggestion. As mentioned by the Reviewer, piezoelectric scaffolds have been shown to support antibacterial capabilities potential and promote bone regeneration¹⁻³. However, the application of piezoelectric scaffolds in bone tissue engineering has

encountered the following challenges: (i) The heterogeneous interface of traditional piezoelectric materials, including piezoelectric polymers and inorganic crystals, are quite different from that of natural bone. During bone growth, the unique chemical composition and hierarchical micro-nanotopology of natural bone have been shown to play crucial roles in the recruitment and differentiation of stem cells. Although piezoelectric materials can activate ion channels through electrical stimulation, their osteoinductive capacity is hardly comparable to that of natural bone interfaces due to the difficulty in simulating the microenvironment of bone in terms of chemical composition and surface topology⁴. (ii) Most piezoelectric materials are highly dependent on the excitation of external energy fields including ultrasonic waves, to perform electrical stimulation, and this reliance on exogenous energy excitation is not conducive to continuous, sequential, uninterrupted treatment. Although it has been reported that some piezoelectric films can generate electrical signals through deformation caused by traction force generated by cells attached to the surface, the accompanying electrical signals tend to be weak due to the small traction force of the cells. In addition, cellular force-driven piezoelectricity is also highly dependent on cell state (e.g. cells must form mature focal adhesions) and susceptible to interference from non-specific adsorption of biomacromolecules^{5,6}. Therefore, implementing long-acting and effective electrical stimulation is also a huge challenge. (iii) strong piezoelectric effect may inhibit cell activity and is not conducive to the osteogenic differentiation of stem cells^{7,8}. (iv) Finally, to meet the requirements of bone regeneration *in vivo*, the biosafety and degradability of piezoelectric biomaterials also need to be carefully examined. However, achieving a balance between degradation and bone regeneration remains a challenge for conventional piezoelectric biomaterials³.

To address the above issues, in our study, we fabricated a self-promoted electroactive mineralized scaffold (sp-EMS) by biomimetic self-assembly of mineralized collagen fibers and ultrathin silver nanowires to treat infected bone defects. Compared with conventional piezoelectric biomaterials, the sp-EMS has the following advantages: (i) The flexibility and spatial configurational freedom of ultrathin silver nanowires allowed them to be coassembled inside the sp-EMS fibers, instead of being exposed to the surface. The exposed surface of the sp-EMS was highly biocompatible mineralized collagen, similar to the chemical composition and micro-nanotopology of natural bone. The bone-like microenvironment of the sp-EMS was favorable for stem cell recruitment and osteogenic differentiation. (ii) The sp-EMS could self-promote the sustained release of weak electric current through mild electrochemical reaction. Therefore, the self-promoted electrical stimulation based on the sp-EMS does not rely on external excitation sources, and will not be interfered by the cell adhesion state and non-specific adsorption of biomacromolecules. (iii) The sp-EMS produced mild electrical stimulation, which promoted the proliferation and differentiation of stem cells. (iv) The sp-

EMS had suitable biodegradability. The animal experiments showed that the sp-EMS gradually degraded after 2 weeks of self-promoted electrical stimulation, and completely degraded at 12 weeks when bone defects were healed. These findings indicate that the sp-EMS could achieve a good balance between material degradation and bone regeneration. As suggested, the advantages of the sp-EMS mentioned above have been included in the Discussion section of the revised manuscript to highlight the novelty (Page 18, Line 551-574; Page 19, Line 592-610, highlighted by yellow).

Q2: the long-term potential issues related to the used of nanoparticles must be addressed (e.g. secondary reactions based on the large surface area and small dimensions, able to interact to cell components).

Reply: We appreciate the Reviewer's comment. In our work, the ultrathin nanowires were not in direct contact with cells. Due to the flexibility and spatial configurational freedom of the ultrafine silver nanowires, they were assembled inside the sp-EMS fibrils along the extension direction of the sp-EMS fibrils during the co-assembly process. The main component exposed on the surface of the sp-EMS and in direct contact with cells is mineralized collagen, which is the reason why the sp-EMS is able to maintain good biocompatibility. It is worth noting that although silver nanowires did not directly interact with cellular components, the sp-EMS could release silver ions due to electrochemical reactions that occur during self-promoted electrical stimulation. It has been confirmed that silver ions could diffuse to proximal as well as distal tissues after implantation into the body, which can be eliminated through the liver and kidneys, and finally excreted through urine and feces⁹. In our original manuscript, we have shown that the biosafety of the sp-EMS both in calvaria and major metabolic organs (liver and kidney) after 12 weeks of implantation by SEM and HE staining (Revised Supplementary Fig. 9a and 9d). As suggested by the Reviewer, in the revised manuscript, we have added more evidences of long-term safety assessment after sp-EMS implantation to address concerns about potential issues. The inductive coupled plasma emission spectrometer (ICP-OES) also confirmed that after 12 weeks of implantation, there was almost no residual silver in major metabolic organs, consistent with the silver levels in unexposed subjects¹⁰. The silver ions released by the sp-EMS could be eliminated from the body through urine and feces (Revised Supplementary Fig. 9b). The detailed information has been included in the Results section in the revised manuscript (Page 11, Line 320-328, highlighted by yellow)

Revised Figure in Supplementary information:

Revised supplementary Fig. 9. The biosafety the sp-EMS *in vivo*. **a**, (i) A representative SEM image and mapping of newly-formed calvaria after implantation of the sp-EMS for 12 weeks. (ii) Weight content of C, N, O, Ca, and P in newly-formed calvaria. **b**, (i) The ICP-OES test of silver concentration in rat urine and feces after implantation of the sp-EMS for 2 weeks. (ii) The ICP-OES test of silver concentration in calvaria, liver, and kidney after implantation of the sp-EMS for 12 weeks. **c,d**, Representative HE staining images of main metabolic organs (liver and kidney) after the implantation of different scaffolds at 2 weeks (c) and 12 weeks (d).

Q3: Ag and Ag related reactions can have negative effects over time, once in the body. The authors must elaborate on that.

Reply: We appreciate the Reviewer's comment. In our original manuscript, we have shown that the biosafety of the sp-EMS both in calvaria and major metabolic organs (liver and kidney) after 12 weeks of implantation by SEM and HE staining (Revised Supplementary Fig. 9a and 9d). According to the reviewer's suggestion, we have supplemented related experiments to systematically assess the potential effects that Ag and Ag related reactions may have over time. The ICP-OES confirmed that after 12 weeks of implantation, the silver ion concentration in major metabolic organs was very low, consistent with the silver levels in unexposed subjects. The silver ions released by the sp-EMS could

be eliminated from the body through urine and feces (Revised Supplementary Fig. 9b). It is also worth noting that the silver ion concentration in the organ is far below the critical concentration that is harmless to the human body stipulated by the World Health Organization [0.1 ppm (100 ppb) in drinking water]¹¹. It has been reported that there were no significant changes in rat kidney, spleen, lungs, heart, testis, and brain with 5 mg/kg b.w of Ag nanoparticles. However, 10 mg/kg b.w of Ag nanoparticles showed moderate degree of cell swelling and vacuolar degeneration in liver¹². In our study, rats, rabbits, and beagle dogs were all exposed to 0.55 mg/kg b.w Ag nanowires, which was far below the toxic level. Moreover, HE staining confirmed that both short-term (2 weeks) and long-term implantation of sp-EMS (12 weeks) showed no obvious toxicity to major metabolic organs (Revised Supplementary Fig. 9c and 9d). Taken together, we have demonstrated that the silver component in the sp-EMS is metabolizable without causing accumulation, and does not have significant negative effects on tissues and organs over time. Of course, we will conduct more long-term testing before clinical use of the sp-EMS. The detailed information has been included in the Results section in the revised manuscript (Page 11, Line 320-328, highlighted by yellow).

Reviewer #2

General Comment: *The work will be of significance to the field bone tissue regeneration and development of electroactive bone graft materials. The synthesis of Ag nanowire, bactericidal effects/osteoinductive property of Ag and the implementation in biomaterials for bone regeneration is not entirely novel. However, the determination of micro-current that is self-generated by electrochemical reaction of Ag nanowires occurring in the scaffold would be the unique aspect of this manuscript.*

Furthermore, the authors have also performed a rather all-rounded investigation; both in vitro and in vivo. Specifically, three different animal models are used and comparable results are documented. Their hypotheses are well supported with the presented results and have filled up some current gaps-in-knowledge in the field

The noteworthy results are: 1) Microcurrent of 4 μ A are generated spontaneously within the sp-EMS, which played a significant role in enhancing the bactericidal effects, angiogenic/osteoinductive properties of the scaffold; 2) Comparable results are documented in the in vitro experiments with stem cells and further validated in three different animal models.

The majority of work performed in this study has supported the conclusions and claims. The data analysis and interpretation are nicely done and clearly presented. The methodologies implemented in this study are sound and well-designed with accurate controls and sufficient details are provided by the authors. They meet the expected standards and are sometimes above expectations. A job well done by the authors.

Reply: We thank the Reviewer for these very positive comments. As suggested, we have made changes in these sections as follows.

Q1: *The results clearly show the bactericidal effects and osteoinductive properties of sp-EMS (Fig 3, 4 and 5). Could the authors further commend the effects of sp-EMS on the inflammatory responses/macrophage activation at the site of critical defects in the animal models?*

Reply: Thank you for your professional comments and valuable suggestions. To better understand the effects of the sp-EMS on the inflammatory responses and macrophage activation during infected bone regeneration, we have included immunofluorescence staining in defect areas after 2 weeks of implantation (Revised Fig. 5e, 5f, and 5g; Revised Supplementary Fig. 13). The percentage of TNF- α^+ cells in the sp-EMS group (2.97% \pm 2.23%) was much lower than that in the MS (48.59% \pm 7.83%) and blank groups (58.68% \pm 3.85%, $p < 0.001$). The same tendency was detected in the percentage of iNOS $^+$ cells. In addition, much more CD68 $^+$ CD163 $^+$ M2 macrophages (64.90% \pm

4.06%) were observed in the sp-EMS group than those in the MS (11.16% ± 1.06%) and blank groups after 2 weeks of implantation (5.84% ± 0.73%, $p < 0.001$), whereas the number of CD68⁺iNOS⁺ M1 macrophages (6.92% ± 2.97%) was much lower than that in the MS (41.18% ± 7.39%) and blank groups (49.72% ± 5.00%, $p < 0.001$). These results indicated that implantation of the sp-EMS into the infected calvarial defects provokes M2 macrophage polarization, which alleviates inflammation status and promotes the bacteria-infected bone regeneration. As suggested, we have included more detailed information in the revised Results section (Page 14-15, Line 446-459, highlighted by yellow).

Revised Figure in main text:

Revised Fig. 5. The sp-EMS achieves nearly complete *in situ* bone regeneration in rats with single bacterial infection. a, Schematic of the healing process of rat infected calvarial bone defects after the cell-free sp-EMS implantation. **b**, Bacteria culture of tissues after different scaffold implantation for 1 week. **c**, (i) Representative μ CT images of rat infectious calvarial bones in

different groups after 12-week implantation. Yellow circles display defect boundary. (ii) Semiquantification of the bone volume and the ratio of new bone to total areas in (i). $n = 5$. ***: $P < 0.001$ versus blank; ###: $P < 0.001$ versus MS. **d**, Representative μ CT, HE, and Masson's trichrome staining images of the infected engineered bones in different groups after 12-week implantation. Yellow boxed areas show the defect areas. Black boxed areas show high-magnification views. ST: soft tissue, S: scaffolds, NB: new bone, V: vessels, OB: osteoblasts. **e**, Representative immunofluorescence images of TNF- α staining (green) and iNOS staining (red) in defect areas after 2 weeks of implantation. **f**, Representative immunofluorescence images of CD68 staining (green) and iNOS staining (red) in defect areas after 2 weeks of implantation. **g**, Representative immunofluorescence images of CD68 staining (green) and CD163 staining (red) in defect areas after 2 weeks of implantation.

Revised Figure in Supplementary information:

Revised supplementary Fig. 13. Semiquantification of TNF- α^+ (a), iNOS $^+$ (b), CD68 $^+$ iNOS $^+$ (c), and CD68 $^+$ CD163 $^+$ (d) cells in rat infected calvarial bone defect areas after 2-week implantation in Fig. 5. $n = 5$, ***: $P < 0.001$ versus blank and MS.

Q2: Please include live measurements of electric current between the platinum sheets as Supplementary materials. So as to prove that the cells were indeed exposed to 4 μ A and there were no current depletion in the well.

Reply: Thank you for your professional comments and valuable suggestions. We have included the movie of exogenous electrical stimulation to BMSCs, which confirmed no current depletion in the well (Movie R1).

Movie R1. Video of exogenous electrical stimulation to BMSCs.

Q3: The interpretation of relative gene expression with only GAPDH as the sole housekeeping gene could be less accurate. A recent study shows fluctuation in GAPDH after electrical stimulation and PPIA, YWHAZ are recommended (doi: <https://doi.org/10.3390/app12010153>). Nevertheless, additional endogenous control should at least be used with GAPDH and is recommended in future studies.

Reply: Thanks for your kind reminder. As suggested, additional endogenous controls (PPIA and YWHAZ) were used for the interpretation of relative gene expression. The expression levels of these two genes relative to GAPDH are similar (Revised supplementary Fig. 16). The detailed information has been included in the Methods section in the revised manuscript (Page 25-26, Line 816-819,

highlighted by yellow). We will choose at least two additional endogenous controls with GAPDH in our future studies.

Revised Figure in Supplementary information:

Revised supplementary Fig. 16. Relative mRNA expression levels of PPIA and YWHAZ normalized to GAPDH in BMSCs after electrical stimulation for 48 h.

Q4: The title is rather general the “Mineralized Scaffold” should be better defined. Perhaps “regeneration of bacteria infected bone defects” is a more appropriate term to use at the end.

Reply: We appreciate the Reviewer’s comment. In this work, we constructed a self-promoted electroactive scaffold with a bone-like interface through a biomimetic self-assembly strategy. The sp-EMS achieves complete or nearly complete *in situ* infected bone healing, from a rat calvarial defect model with single bacterial infection, to a rabbit open alveolar defect model and a beagle dog vertical bone defect model with the complex oral bacterial microenvironment. In view of the above characteristics, we fully accept the Reviewer’s suggestion and have changed the title as “Self-promoted electroactive biomimetic mineralized scaffolds for bacteria-infected bone regeneration” (Page 1, Line 1-2, highlighted by yellow).

Q5: “Infected bone loss” in line 25 is probably not a correct term, do the authors mean “Infected bone defect” or “Infection driven bone loss”?

Reply: We appreciate the Reviewer's constructive comments. We have corrected the description of “infected bone loss” to “infected bone defect” (Page 2, Line 27, highlighted by yellow).

Q6: Please check the references cited, as the citations do not explain the “Statements” made. For example:

a. Ref 1,2 (line 45): High treatment failure rate of infected bone defects are stated, but literatures for drug-eluting joint implant and modifies antibacterial TiO2 implants that showed improved treatment outcomes are cited.

Reply: We apologize for this error. We have corrected the reference in the revised manuscript (Page 3, Line 48; Page 34, Line 1082-1086, highlighted by yellow).

b. Ref 9,10 (line 58-59): The term “Bioelectricity” is used. This encompasses dielectric, piezoelectric, pyroelectric, ferroelectric, streaming potential properties. Both citations are only referring to piezoelectric properties of tissues. Additionally “maintaining the physiological function of bone” is stated, but Ref10 mentioned only synthetic piezoelectric electrospun fibers. Do not see the link.

Reply: We appreciate the Reviewer’s suggestion. We apologize for the ambiguity caused. We used the term “bioelectricity” to refer to the inherent electrical properties of natural bone, known as piezoelectricity. For natural bone, pyroelectric and ferroelectric effects are negligible. That is why we only cite references on the piezoelectric properties of tissues. To avoid ambiguity, we have replaced “bioelectricity” with “the piezoelectricity of natural bone”. Additionally, we have also cited references on the critical role of natural bone piezoelectricity for maintaining physiological function. We have corrected the references in the revised manuscript (Page 3, Line 62; Page 34, Line 1105-1113, highlighted by yellow).

c. Ref 16 (line 70): self-polarization process of bone is stated in line 69, but Ref 16 is a study on modified titanium surface.

Reply: We apologize for the error that occurred when inserting references. Actually, electrical stimulation has been proven to promote osteoblast attachment and proliferation while inhibiting

bacterial activity. We have corrected the sentence and changed references in the revised manuscript (Page 3, Line 72-73; Page 35, Line 1129-1131, highlighted by yellow).

d. Ref 18 (line 101): It is stated that “Young’s modulus of Ag uNW ranged from 0.22-0.59 GPa was similar to that of collagen fibrils”. However, nowhere in the cited literature had listed these values. The lowest Young’s modulus for EMC was 1.01 +/- 0.24 GPa. Also, are the authors referring to mineralized collagen fibrils? Please clarify.

Reply: We thank the Reviewer for pointing out this issue. We referred to the Young's modulus of Ag uNWs similar to that of unmineralized collagen microfibrils with 0.3–1.2 GPa¹³. We have changed the reference in the revised manuscript (Page 5, Line 114; Page 35, Line 1141-1143, highlighted by yellow).

Q7: The term “inflammatory tissues” (line 48) is perhaps misused? Is inflammatory response or inflamed tissues referred?

Reply: We thank the Reviewer for this kind reminder. We have corrected the description of “inflammatory tissues” to “infected tissues” (Page 3, Line 51, highlighted by yellow).

Q8: Antibiotics... “destroyed the body’s microbial and immune environment,...”(line 54), do the authors mean antimicrobial environment?

Reply: We thank the Reviewer for the constructive comment. We mean that antibiotics disrupt the balance of the body’s microbial ecosystem, thereby suppressing the osteoinductive properties of bone grafts. To make the statement clearer, we have modified the description in the revised manuscript (Page 3, Line 56-57, highlighted by yellow).

Q9: “shear-force of collagen” line 60 refers to piezoelectric property of bone tissue upon application of mechanical force? The sentence is overly simplified, not sure what the authors mean here.

Reply: We appreciate the Reviewer’s comment. We have corrected the sentence to “Bone tissue can spontaneously generate endogenous electrical signals, which are derived from the polarization of collagen fibrils caused by mutual misalignment under the applied shear force.” (Page 3, Line 63-64, highlighted by yellow).

Q10: The term “almost dead” in line 357 seems inappropriate here, since there are no means of measurement to determine that.

Reply: We thank the Reviewer’s suggestion. The term “almost dead” comes from the results of live/dead staining. The vast majority of *S. aureus* co-cultured with the sp-EMS were labeled with PI-DNA (capable of penetrating the cell membrane and intercalating into the bacterial DNA double helix, producing a red fluorescent signal), indicating that most of the bacteria were dead ($74.53\% \pm 4.06\%$). In contrast, the proportions of bacteria labeled by Calcein-AM in the MS group and blank groups were $96.68\% \pm 0.16\%$ and $98.19\% \pm 1.17\%$, respectively, suggesting that the bacteria in these two groups remained viable. To make this statement clearer, the detailed information has been included in the revised manuscript (Page 12, Line 372-376, highlighted by yellow).

Reviewer #3

Q1: Rephrase this sentence, as it is not really clear and I do not know what latent reproduction is supposed to mean: The open environment of bone defects is conducive to the latent reproduction of various pathogenic bacteria.

Reply: We thank the Reviewer's comment. Due to the open bacterial microenvironment in oral cavity, there are large quantities of opportunistic pathogens including aerobic bacteria, anaerobic bacteria and facultative anaerobic bacteria, which colonize on the surface of soft and hard tissues in oral cavity. The latent reproduction means the incubation and repeated stimulation of pathogenic bacteria, which may lead to infected bone defects. To make this statement clear, we have revised the sentence as follows: "The open environment of bone defects in oral cavity is conducive to the colonization and reproduction of various opportunistic pathogens". The detailed information has been included in the revised manuscript (Page 3, Line 49-50, highlighted by yellow).

Q2: This sentence should also be edited for clarity. Perhaps it may be split into two sentences, and it is not really correct to say antibiotics destroy the immune environment. Please adapt the sentence to have only correct statements: However, it is difficult for antibiotics to deal with complex infections caused by drug resistant bacteria; more seriously, antibiotics also inhibit the osteoinductive properties 54 of bone grafts by destroying the body's microbial and immune environment, ultimately leading to a "double loss" of anti-infection and bone repair.

Reply: We appreciate the reviewer's comment. As suggested, we have revised the sentence as follows: "However, it is difficult for antibiotics to deal with complex infections caused by drug-resistant bacteria. More seriously, antibiotics also inhibit the osteoinductive properties of bone grafts by destroying the balance of microbial ecosystems, ultimately leading to a "double loss" of anti-infection and bone repair." (Page 3, Line 55-58, highlighted by yellow).

Q3: I would like the introduction to provide more background information on the materials developed. What was characterised and tested prior to this paper?

Reply: We appreciate the Reviewer's comment. It has been demonstrated in our previous studies that the biomimetic self-assembly strategy can thermodynamically control the hierarchical arrangement of hydroxyapatite nanocrystals and collagen molecules to form a biomimetic mineralized collagen interface that is highly similar to natural bone in terms of mechanical

properties, chemical composition, surface topology, biocompatibility, and biodegradability. This bone-like interface can regulate the fate of stem cells and exhibit excellent bone regeneration potential^{14,15}. Inspired by this, further coassembly with electroactive nanomaterials will not only preserve the biomimetic bone-like interface, but also endow the scaffold with antibacterial properties and stronger osteoinductive capabilities. In the present study, the sp-EMS continuously generated a weak current through a mild electrochemical reaction, which stimulated the recruitment of BMSCs and promoted their osteogenic differentiation (Fig. 2). Meanwhile, electrical stimulation also synergized with the electrochemical oxidation product, silver ions, to destroy the cell wall of *S. aureus* and induce oxidative stress, resulting in bacterial death (Fig. 4; Supplementary Fig. 12). As suggested, we have included more information on material development in the Introduction section (Page 4, Line 80-89, highlighted by yellow).

Q4: The DF approach for group size seems to suggest a low number of animals per group. This is not a very common approach. Does a group of 3 animals allow a robust evaluation of efficacy of the treatments?

Reply: We appreciate the Reviewer's comment. The methodological data of sample size calculation in animal experiments were retrieved systematically¹⁶⁻¹⁸. Basically, there are two methods of sample size calculation in animal studies. The most favored and scientific method is calculation of sample size by power analysis. However, to calculate the sample size by power analysis, we must have prior knowledge and information about two major concepts: (i) effect size (the minimum difference between two groups that can be considered clinically significant) and (ii) standard deviation (the measure of variability within a sample for a quantitative variable). Since our study is the first of its kind in dogs, it is not possible to assume about effect size/ standard deviation as no previous findings are available. Therefore, an alternative to the power analysis approach, the 'resource equation' approach, which sets the acceptable range of degrees of freedom (DFs), was used to determine the sample size^{17,19}. We calculated the minimum sample size required and got 3 animals per group, which also follows the principles of the 3Rs (Reduction, Refinement, and Replacement) in the calculation of the sample size in animals. We know that the sample size is small and prone to false negatives. In our study, as a further exploration of the previous partial results in animals such as rats, the partial results provide evidence support for large animals to a certain extent. However, according to the Reviewer's suggestion, in order to facilitate readers' understanding, we have added a corresponding limitation in the Methods section (Page 29, Line 920-928, highlighted by yellow). The calculation of sample size by power analysis based on our data would be carried out in future studies.

Q5: In any case, I would like to see some data table listing all animal experiments, all groups per experiment and group size per experiment.

Reply: We appreciate the Reviewer’s comment. As suggested, we have listed all animal experiments, all groups per experiment and group size per experiment in the table in Supplementary Information (Revised Supplementary Table 4).

Revised Table in Supplementary Information:

Supplementary Table 4. List of animals used in the study.

Defects	Sample size	Time points	Groups
Rat critical-sized noninfected calvarial defects	45	2 weeks 8 weeks 12 weeks	Blank ($n = 5$ at each time point) MS ($n = 5$ at each time point) sp-EMS ($n = 5$ at each time point)
Rat critical-sized infected calvarial defects	33	1 week 2 weeks 12 weeks	Blank ($n = 5$ at 2 and 12 weeks; $n = 1$ at 1 week) MS ($n = 5$ at 2 and 12 weeks; $n = 1$ at 1 week) sp-EMS ($n = 5$ at 2 and 12 weeks; $n = 1$ at 1 week)
Rabbit open bone defects	5 (bilateral alveolar open defects in one rabbit and totally 10 defects)	8 weeks	Blank ($n = 3$) MS ($n = 3$) sp-EMS ($n = 3$)
Dog vertical bone defects	3 (two bone defects at each side and four defects in one dog and totally 12 defects)	12 weeks	Blank ($n = 3$) Bio-Oss ($n = 3$) MS ($n = 3$) sp-EMS ($n = 3$)

Q6: Since animal welfare is such an important issue, the authors are encouraged to follow best practice and disclose all relevant data pertaining to animal welfare.

The number of animals used per group per model should be disclosed. How was over anaesthesia performed? How many animals were included but did not survive the surgery or recovery period? How was postoperative pain managed and scored? What were criteria for animal pain or burden to warrant intervention or early euthanasia? If the surgical procedures have not been described before, some details need to be given. For example, saying " Critical-sized calvarial bone defects of 5 mm in diameter were created and different 873 scaffolds were implanted" is not detailed enough to aid the reader who may want to perform this model. So add a reference for a full description, or provide more details here. Was any perioperative antibiotic prophylaxis done in these animals?

Reply: We appreciate the Reviewer's comments. As suggested, the number of animals used per group per model was added in Supplementary Table 4. According to Canadian Council on Animal Care (CCAC) and American Veterinary Medical Association (AVMA) euthanasia guidelines, rats were received 200 mg/kg intraperitoneal injection of sodium pentobarbital for euthanasia. Chemical euthanasia with an overdose (400 mg/kg) of intravenous pentobarbital for rabbits and beagle dogs is chosen for euthanasia. All animals survived the surgery and recovery period. The postoperative pain was scored by grimace scale, a facial expression-dependent measure developed for quantifying spontaneous pain based on human facial coding scales²⁰. When the pain score >2, the animal would be carefully examined. If the score >4, additional intervention should be applied to the animals.

During experiment procedures, rats were initially anesthetized by intraperitoneal injection of 1% pentobarbital sodium solution. After separating the skin and muscle, the calvarium was exposed. The whole layer of bone tissue was removed with a dental implanter (W&H Implantedmed, Austria) at 1200 rpm/min and critical-sized calvarial bone defects of 5 mm in diameter were created. Then different scaffolds were implanted into the defects and the skin was sutured with 6-0 surgical sutures. No perioperative antibiotic was used in these animals. Rats were randomly separated into three groups ($n = 5$): (i) blank group (untreated), (ii) MS group (implanted with the MS) and (iii) sp-EMS group (implanted with the sp-EMS). As suggested, the detailed information has been included in the Methods section in the revised manuscript (Page 29, Line 937-950; Page 30, Line 980; Page 31, Line 994-995, highlighted by yellow).

Q7: Details about how electrodes were applied to the animals, how long they were applied, and how the animal tolerated the electrodes should be given.

Reply: We appreciate the Reviewer's careful reading. To test whether self-promoted electrical stimulation accompanies the bone regeneration process, we measured the potential and current of sp-EMS *in situ* without any exogenous batteries at different periods after implantation using the

electrometer (Keithley 6517B) and oscilloscope (Teledyne LeCroy HD 4096) (Supplementary Movie 1). The current was measured by placing the positive electrode of the electrometer in the scaffolds and the negative electrode at the calvarial bone away from the scaffolds for 5 minutes. As suggested, we have added the details in the Methods section in the revised manuscript (Page 30, Line 954-959, highlighted by yellow).

Q8. Please provide more details for this process: Tissues in the defect area were obtained and placed into 5 mL of sterile LB agar medium. After co-cultured for 24 h, 50 µL of bacterial suspension was coated on solid medium after diluted with PBS. Then, the culture dishes were placed in a bacterial incubator at 37 °C for 24 h and removed to examine the bacterial colony.

Why was the sample placed in LB medium? As it is written I have concerns that the method is not appropriate

Reply: We apologize for the writing error. The LB agar medium was used for the culture for *S. aureus*, and tissues obtained were firstly placed into 5 mL of sterile LB liquid medium. Actually, the sterile LB liquid medium was used for CFU assay. As suggested, the detailed information has been included in the Methods section in the revised manuscript as follows: Tissues in the defect area were obtained and cultured in 5 mL of sterile LB liquid medium for 24 h. After diluting 1000 times with PBS, 50 µL of diluted bacterial suspension was coated on LB agar plates, and cultured at 37 °C for 24 h to examine the bacterial colony (Page 30, Line 966-968, highlighted by yellow).

Q9. For this text, is this 10 per animal? Totally 10 alveolar bone defects were created, which was randomly divided into three groups.

The CT images of the bones may be better as 3D reconstructions instead of individual slices as this will show the entire bone.

Reply: We appreciate the Reviewer's comments. 5 rabbits and totally 10 alveolar bone defects were created and randomly divided into three groups (Revised Supplementary Table 4). As suggested, the 3D reconstructions showed the entire bone regeneration and the defect boundaries, which have been included in Revised Supplementary Fig. 14. To better show the regeneration of cortical bone and cancellous bone, we chose 2D reconstruction slices as the representative images of the regeneration of rabbit alveolar bone defect. The detailed information has been included in the Results section in the revised manuscript (Page 15, Line 472, highlighted by yellow).

Revised Figure in Supplementary Information:

Revised Supplementary Fig. 14. Representative μ CT 3D reconstruction images of the rabbit engineered bones in the blank, MS, and sp-EMS groups after 8 weeks of implantation. White lines display the defect boundaries.

Q10. The methods for the in vitro antimicrobial tests should be substantially clarified and improved: For CFU assay, the scaffolds and the bacteria suspension were co cultured for 24 h. Then 10 mL of sterile LB agar medium was poured into a petri dish for the next tests. According to the manufacturer's instructions, 50 μ L of bacterial suspension was coated on solid medium by a Whitley Automated Spiral plate.

As it is written I have concerns that the method is not appropriate

Reply: We appreciate the Reviewer's suggestion. Actually, the sterile LB liquid medium was used for CFU assay. We have corrected the sentence to "For CFU assay, the scaffolds and the bacteria of 3×10^8 CFU/mL were cocultured for 24 h. Then 10 μ L of suspension was diluted 1000 times with sterile LB liquid medium for the next test. According to the manufacturer's instructions, 50 μ L of diluted bacterial suspension was spread on the solid LB agar plates by a Whitley Automated Spiral plater (Interscience, USA). After incubation at 37 °C for 24 h, the plates were removed to count the number of colonies". As suggested, the detailed information has been included in the Methods section in the revised manuscript (Page 27, Line 870-875, highlighted by yellow).

References:

- 1 Tang, B. *et al.* Harnessing cell dynamic responses on magnetoelectric nanocomposite films to promote osteogenic differentiation. *ACS Appl Mater Interfaces* **10**, 7841-7851, doi:10.1021/acsami.7b19385 (2018).
- 2 Zhu, M. *et al.* Photo-responsive chitosan/Ag/MoS₂ for rapid bacteria-killing. *J Hazard Mater* **383**, 121122, doi:10.1016/j.jhazmat.2019.121122 (2020).
- 3 Liu, Z. *et al.* Electroactive biomaterials and systems for cell fate determination and tissue regeneration: Design and applications. *Adv Mater* **33**, e2007429, doi:10.1002/adma.202007429 (2021).
- 4 Woodruff M. A *et al.* Bone tissue engineering: from bench to bedside. **15**, doi:10.1016/S1369-7021(12)70194-3 (2012).
- 5 Hwang, G. T. *et al.* Flexible piezoelectric thin-film energy harvesters and nanosensors for biomedical applications. *Adv Healthc Mater* **4**, 646-658, doi:10.1002/adhm.201400642 (2015).
- 6 Liu, Z. R. *et al.* Cell-traction-triggered on-demand electrical stimulation for neuron-like differentiation. *Adv Mater* **33**, 2106317, doi:<https://doi.org/10.1002/adma.202106317> (2021).
- 7 Li, H. *et al.* Enhanced ferroelectric-nanocrystal-based hybrid photocatalysis by ultrasonic-wave-generated piezophototronic effect. *Nano Lett* **15**, 2372-2379, doi:10.1021/nl504630j (2015).
- 8 Kapat, K. *et al.* Piezoelectric nano-biomaterials for biomedicine and tissue regeneration. *Adv Funct Mater* **30**, 1909045, doi:10.1002/adfm.201909045 (2020).
- 9 Hadrup, N. *et al.* Distribution, metabolism, excretion, and toxicity of implanted silver: a review. *Drug Chem Toxicol* **45**, 2388-2397, doi:10.1080/01480545.2021.1950167 (2022).
- 10 Armitage, S. A. *et al.* The determination of silver in whole blood and its application to biological monitoring of occupationally exposed groups. *Ann Occup Hyg* **40**, 331-338, doi:10.1016/0003-4878(95)00076-3 (1996).
- 11 *WHO Guidelines Approved by the Guidelines Review Committee.* (World Health Organization Copyright © World Health Organization 2017).
- 12 Nakkala, J. R. *et al.* Green synthesized silver nanoparticles: Catalytic dye degradation, in vitro anticancer activity and in vivo toxicity in rats. *Mater Sci Eng C Mater Biol Appl* **91**, 372-381, doi:10.1016/j.msec.2018.05.048 (2018).
- 13 Gautieri, A. *et al.* Hierarchical structure and nanomechanics of collagen microfibrils from the atomistic scale up. *Nano Lett* **11**, 757-766, doi:10.1021/nl103943u (2011).

- 14 Liu, Y. *et al.* Hierarchically staggered nanostructure of mineralized collagen as a bone-grafting scaffold. *Adv Mater* **28**, 8740-8748, doi:10.1002/adma.201602628 (2016).
- 15 Liu, Y. *et al.* Thermodynamically controlled self-assembly of hierarchically staggered architecture as an osteoinductive alternative to bone autografts. *Adv Funct Mater* **29**, 1806445, doi:10.1002/adfm.201806445 (2019).
- 16 Liu, Y. *et al.* Exercise-induced piezoelectric stimulation for cartilage regeneration in rabbits. *Sci Transl Med* **14**, eabi7282, doi:10.1126/scitranslmed.abi7282 (2022).
- 17 Arifin, W. N. *et al.* Sample size calculation in animal studies using resource equation approach. *Malays J Med Sci* **24**, 101-105, doi:10.21315/mjms2017.24.5.11 (2017).
- 18 Festing, M. F. *et al.* Guidelines for the design and statistical analysis of experiments using laboratory animals. *Ilar j* **43**, 244-258, doi:10.1093/ilar.43.4.244 (2002).
- 19 Mead, R. *et al.* Statistical principles for the design of experiments. doi:10.1017/CBO9781139020879. (2012).
- 20 Nagakura, Y. *et al.* Spontaneous pain-associated facial expression and efficacy of clinically used drugs in the reserpine-induced rat model of fibromyalgia. *Eur J Pharmacol* **864**, 172716, doi:10.1016/j.ejphar.2019.172716 (2019).

REVIEWER COMMENTS

Reviewer #1 (Remarks to the Author):

The authors have performed an excellent work in the revision of the manuscript, taking into consideration all comments from the reviewers and providing suitable answers and modifications to the manuscript. This referee supports publication of the work in the present form.

Reviewer #2 (Remarks to the Author):

The authors did a good job addressing my previous queries and had made the corrections necessary to improve clarity.

I recommend to publish this manuscript at its current form and wish the authors all the best on further research in this area.

Reviewer #3 (Remarks to the Author):

The authors have prepared a substantial response to reviewer comments.

The responses by the authors are overall quite satisfactory. The edits in the manuscript are comparatively less detailed, but in most cases adequate. I would encourage the authors to provide more edits to the actual manuscript as much of the information in the response document is more detailed than in the manuscript itself.

I only have 2 important points remaining, both of which were flagged in the initial review. It relates to the processing of the bacteriological assays. The responses by the authors confirm my concerns, and I consider the test result (at least in vivo) to be invalid.

1. For the in vitro test, the fluid/media used to suspend the bacterial culture is not mentioned. Was it PBS, of LB, or something else?
2. For the in vivo tests, the tissues from the animal was incubated in LB medium for 24 hours. THs will result in the bacteria in the tissue multiplying, and so the eventual CFU result is not reflective of in vivo activity, but rather of regrowth in LB medium. Therefore, the result is not valid. THs experiment needs to be removed from the manuscript in my opinion.

Responses to comments from the Reviewers

Reviewer #1

General Comment: The authors have perform an excellent work in the revision of the manuscript, taking into consideration all comments from the reviewers and providing suitable answers and modifications to the manuscript. This referee supports publication of the work in the present form.

Reply: We thank the Reviewer for this very positive comment.

Reviewer #2

General Comment: The authors done a good job addressing my previous queries and had made the corrections necessary to improve clarity. I recommend to publish this manuscript at its current form and wish the authors all the best on further research in this area.

Reply: We thank the Reviewer for this very positive comment.

Reviewer #3

General Comment: The authors have prepared a substantial response to reviewer comments. The responses by the authors are overall quite satisfactory. The edits in the manuscript are comparatively less detailed, but in most cases adequate. I would encourage the authors to provide more edits to the actual manuscript as much of the information in the response document is more detailed than in the manuscript itself.

Reply: We thank the Reviewer for these very positive comments. As suggested, we have included more details on sample size calculation and animal welfare from the previous response document in the revised Methods section (Page 28, Line 901-915, highlighted by yellow).

Q1: 1. For the in vitro test, the fluid/media used to suspend the bacterial culture is not mentioned. Was it PBS, of LB, or something else?

Reply: We appreciate the Reviewer's comment. In our work, the LB liquid medium was used to suspend bacterial culture. As suggested, the detailed information has been included in the Methods section of the revised manuscript (Page 26-27, Line 851-852, highlighted by yellow).

Q2: For the *in vivo* tests, the tissues from the animal was incubated in LB medium for 24 hours. THs will result in the bacteria in the tissue multiplying, and so the eventual CFU result is not reflective of *in vivo* activity, but rather of regrowth in LB medium. Therefore, the result is not valid. THs experiment needs to be removed from the manuscript in my opinion.

Reply: Agree. As suggested, we have removed the CFU result in the revised manuscript (Revised Fig. 5). To demonstrate the establishment of an infectious bone defect model, we have included HE staining images of the blank group in the revised Supplementary Fig. 13. Compared to the rat calvarial noninfected bone defect model, the infected bone defect model without any implants exhibited a large accumulation of inflammatory cells 2 weeks after surgery, confirming the presence of infection (Revised Supplementary Fig. 13). The detailed information has been included in the revised supplementary file (Page 14, Line 417-420, highlighted by yellow).

Revised Figure in main text:

Revised Fig. 5. The sp-EMS achieves nearly complete *in situ* bone regeneration in rats with single bacterial infection. **a**, Schematic of the healing process of rat infected calvarial bone defects after the cell-free sp-EMS implantation. **b**, (i) Representative μ CT images of rat infectious calvarial bones in different groups after 12-week implantation. Yellow circles display defect boundary. (ii) Semiquantification of the bone volume and the ratio of new bone to total areas in (i). $n = 5$. ***: $P < 0.001$ versus blank; ####: $P < 0.001$ versus MS. **c**, Representative μ CT, HE, and Masson's trichrome staining images of the infected engineered bones in different groups after 12-week implantation. Yellow boxed areas show the defect areas. Black boxed areas show high-magnification views. ST: soft tissue, S: scaffolds, NB: new bone, V: vessels, OB: osteoblasts. **d**, Representative immunofluorescence images of TNF- α staining (green) and iNOS staining (red) in defect areas after 2 weeks of implantation. **e**, Representative immunofluorescence images of CD68 staining (green) and iNOS staining (red) in defect areas after 2 weeks of implantation. **f**, Representative immunofluorescence images of CD68 staining (green) and CD163 staining (red) in defect areas after 2 weeks of implantation.

Revised Figure in Supplementary information:

Revised Supplementary Fig. 13. Establishment of a rat calvarial infected bone defect model. Compared to the rat calvarial noninfected bone defect model, the infected bone defect model without any implants exhibited a large accumulation of inflammatory cells 2 weeks after surgery, confirming the presence of infection. ST: soft tissue. Green arrow displays inflammatory cells.

REVIEWERS' COMMENTS

Reviewer #3 (Remarks to the Author):

The authors made a clear, constructive change to their manuscript. I am happy that they removed the invalid data. No further issues.

Responses to comments from the Reviewer

Reviewer #3

General Comment: The authors made a clear, constructive change to their manuscript.

I am happy that they removed the invalid data. No further issues.

Reply: We thank the Reviewer for this positive comment.